# Viral subversion of selective autophagy is critical for biogenesis of virus replication organelles

Yun Lan[1,4], Sophie Wilhelmina van Leur[2,4], Julia Ayano Fernando[1], Ho Him Wong[1], Martin Kampmann[1], Lewis Siu[1], Jingshu Zhang[1], Mingyuan Li[1], John M. Nicholls[3] & Sumana Sanyal ◉ [1,2] ✉

Infection by many (+)RNA viruses is accompanied by ER-expansion and membrane remodelling to form viral replication organelles, followed by assembly and secretion of viral progenies. We previously identified that virus-triggered lipophagy was critical for flaviviral assembly, and is driven by the lipid droplet associated protein Ancient ubiquitin protein 1 (Aup1). A ubiquitin conjugating protein Ube2g2 that functions as a co-factor for Aup1 was identified as a host dependency factor in our study. Here we characterized its function: Ube2g2-deficient cells displayed a dramatic reduction in virus production, which could be rescued by reconstituting the wild-type but not the catalytically deficient (C89K) mutant of Ube2g2, suggesting that its enzymatic activity is necessary. Ube2g2 deficiency did not affect entry of virus particles but resulted in a profound loss in formation of replication organelles, and production of infectious progenies. This phenomenon resulted from its dual activity in (i) triggering lipophagy in conjunction with Aup1, and (ii) degradation of ER chaperones such as Herpud1, SEL1L, Hrd1, along with Sec62 to restrict ER-phagy upon Xbp1-IRE1 triggered ER expansion. Our results therefore underscore an exquisite fine-tuning of selective autophagy by flaviviruses that drive host membrane reorganization during infection to enable biogenesis of viral replication organelles.

A universal feature of positive-sense RNA viruses (e.g. flaviviruses and coronaviruses) is that they replicate in the host cell cytoplasm by remodelling intracellular membranes to form virus replication compartments. These are medically relevant human pathogens that are widespread and afflict millions worldwide. For instance, dengue is responsible for about 100 million confirmed cases globally each year[1,2]. Zika virus (ZIKV), a related flavivirus poses an imminent threat with its global spread and associated neurotoxicity[3]. No antivirals exist for either virus, and the efficacy of the available vaccine for dengue is yet to be assessed. Similarly, coronaviruses have emerged as pathogens

with enormous pandemic potential[4]. Elucidation of host cellular pathways that are exploited in the course of infection is therefore necessary to determine fundamental biology underpinning development of therapeutics.

Once internalised through receptor-mediated endocytosis, the positive strand viral RNA genome is delivered to the cytosol[5]. The translated polyprotein is cleaved by host and viral proteases to generate three structural and seven non-structural proteins in case of dengue and Zika. Subsequent replication and assembly of progeny virions require massive rearrangements of intracellular membranes to

[1]HKU-Pasteur Research Pole, School of Public Health, Li Ka Shing Faculty of Medicine, University of Hong Kong, Hong Kong SAR, China. [2]Sir William Dunn School of Pathology, University of Oxford, South Parks Road, Oxford OX1 3RE, UK. [3]Department of Pathology, Li Ka Shing Faculty of Medicine, University of Hong Kong, Hong Kong SAR, China. [4]These authors contributed equally: Yun Lan, Sophie Wilhelmina van Leur. ✉e-mail: sumana.sanyal@path.ox.ac.uk

form replication compartments[3]. These are most likely endoplasmic reticulum (ER) derived[5]. Electron tomography and 3D-reconstructions have provided some insights into the architecture of such virus-induced organelles. Replication compartments are speculated to be sites for concentrating viral proteins and RNA which in turn facilitate assembly and release[6].

Our previous study revealed a critical role of lipid droplet hydrolysis via autophagy (lipophagy) in assembly and secretion of dengue and Zika viral progenies[7,8]. Selective autophagy was crucial not only for lipophagy but also for secretion of a population of virus particles[9,10]. As with all RNA viruses, infection with dengue/Zika is accompanied by massive alteration in cellular ubiquitylation and ubiquitin like modifications, some of it host-driven as part of the immune signalling cascades[11–14] and others virus-driven to trigger cellular reprogramming[8,15]. Changes in ubiquitylation of ER and lipid droplet-associated host factors revealed Aup1 and Ube2g2[8].

Ube2g2, is a E2-conjugating enzyme in the ubiquitylation cascade, reported to function as a co-factor of Aup1[16,17]. In this study we investigated the role of Ube2g2 in the process of virus biogenesis using flavi and coronaviruses. Ube2g2-deleted cells were severely impaired in production of viral progenies, which could be rescued in cells reconstituted with the wild-type but not the catalytically inactive mutant of Ube2g2. Interestingly, Ube2g2 was found to not only function in conjunction with Aup1 to trigger lipophagy, but also inhibited ER-phagy to

enable membrane remodelling during virus replication. Consequently, the effect of Ube2g2 deficiency was more pronounced than Aup1 deletion. Loss of Ube2g2 activity resulted in a complete block in ER-remodelling, necessary to form virus replication organelles. Virus replication was therefore significantly impaired with rapid degradation of viral non-structural proteins that form the replication complex. Mechanistically, the viral NS2b-NS3 protease and Ube2g2-dependent degradation of ER-stress chaperones Herp/Herpud1, Calreticulin and Sel1L were found to be critical for inhibiting ER-phagy, which occurred in a Sec62 and Chmp4-dependent manner in Ube2g2-deficient cells to impair biogenesis of virus replication organelles and secretion of progeny virions.

## Results

### Ube2g2-deficient cells are resistant to +RNA viruses

Ube2g2 is a cofactor of Aup1 that we identified as a potential host dependency factor for flavivirus infections[17]. Ube2g2 is a E2-conjugating enzyme of the ubiquitylation cascade with several distinct binding domains for RING E3-ligases, ubiquitin and G2BR[18,19] (Fig. 1a). To study the functional impact of Ube2g2 on virus infection, we generated Ube2g2[−/−] HeLa cells using CRISPR/Cas9 gene editing. Several single clones with no detectable Ube2g2 were identified; multiple clones were tested to display the same phenotype and clone I was used in the experiments described in the following sections

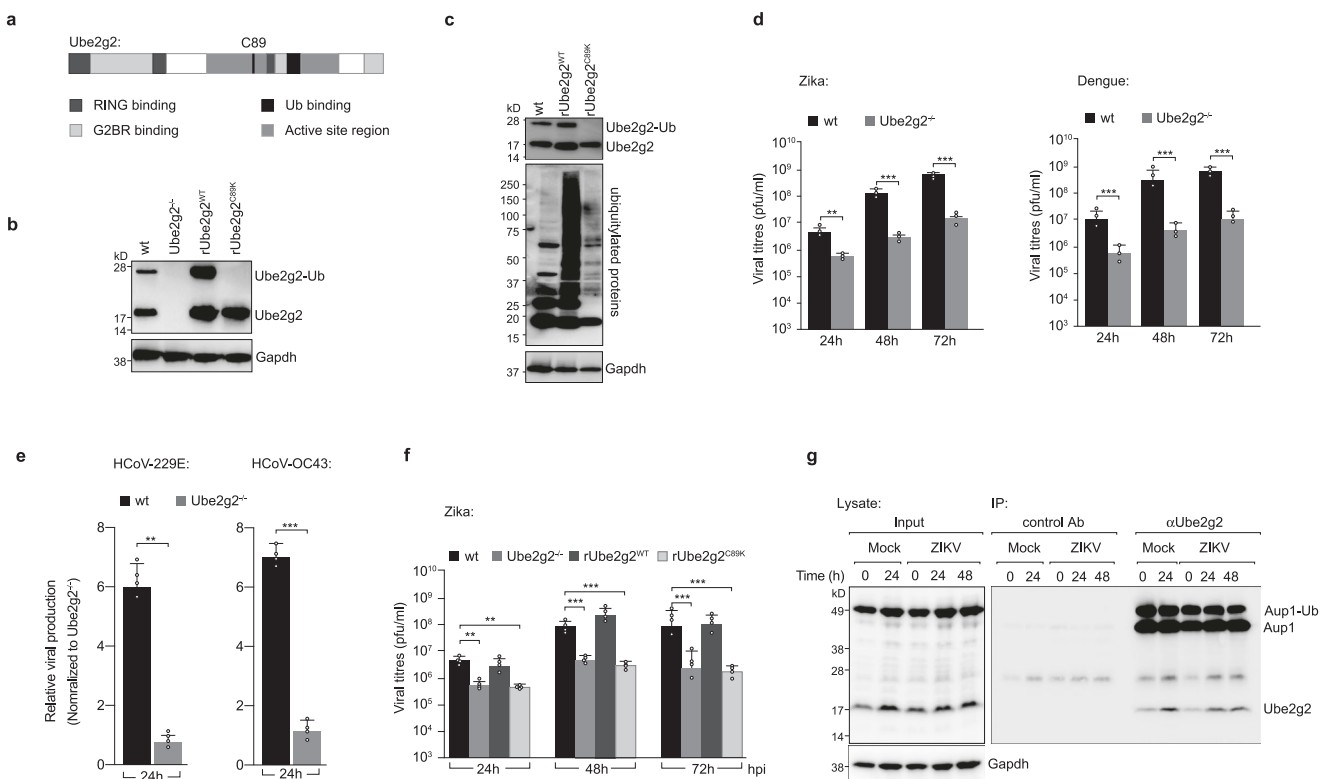

**Fig. 1 | Ube2g2 is necessary for production of infectious virus particles.**
**a** Schematic of the domains in Ube2g2. C89 is the catalytic cysteine mutated to lysine in this study. **b** Single clones of Ube2g2[−/−] HeLa cells were generated using CRISPR/Cas9; a wild-type (rUbe2g2[WT]) and C89K (rUbe2g2[C89K]) mutant variant were reconstituted into the deletion background. Lysates were generated and resolved by gel electrophoresis to determine expression of Ube2g2 by immunoblotting. **c** Cellular ubiquitylation profiles were measured in lysates from rUbe2g2[WT] and rUbe2g2[C89K] cells by immunoblotting. **d** Wild-type and Ube2g2[−/−] cells were infected with ZIKV and DENV at MOI 5. Extracellular virus production at indicated time points were determined by plaque assay. Error bars represent mean ± SD, $n = 3$ biologically independent experiments. **P < 0.01; ***P < 0.001 (compared with wild-type by ANOVA followed by one-sided Dunnett's test). **e** Wild-type and

Ube2g2[−/−] cells were infected with indicated coronaviruses at MOI 0.1. Extracellular virus production at 24 h post infection was determined by qPCR. Error bars represent mean ± SD, $n = 4$ biologically independent experiments. **P < 0.01; ***P < 0.001 (derived using unpaired Student's t test). **f** Wild-type, Ube2g2[−/−], rUbe2g2[WT] and rUbe2g2[C89K] cells were challenged with Zika virus at MOI 5. Infectious virus production was measured using plaque assay. Error bars represent mean ± SD, $n = 4$ biologically independent experiments. **P < 0.01; ***P < 0.001 (compared with wild-type by ANOVA followed by one-sided Dunnett's test). **g** Endogenous Ube2g2 was immunoprecipitated from mock and virus-infected cells at indicated time intervals and probed for Aup1. (Source data are provided as a Source data file).

(Fig. 1b). To further investigate if the catalytic activity of Ube2g2 was essential during Zika virus infection, we generated a cell line stably expressing a catalytically inactive form of Ube2g2 using site-directed mutagenesis. The common catalytic unit of E2 enzymes contains a conserved active-site cysteine (C89), which is essential for Ub/Ubl transfer[18]. We reconstituted Ube2g2[−/−] cells with either the wild-type or catalytically inactive C89K mutants. Both were expressed at comparable levels. While rUbe2g2[WT] was found in its ubiquitin modified form, rUbe2g2[C89K] remained unmodified (Fig. 1b). This was also reflected in total cellular ubiquitylation profiles, which was dramatically reduced in the rUbe2g2[C89K] compared to rUbe2g2[WT] cells (Fig. 1c). Wild-type and Ube2g2[−/−] cells were infected with flaviviruses (Zika or dengue), and viral titres in the supernatants were determined using RT-qPCR and plaque assays. Deletion of Ube2g2 resulted in significant reduction of virus production (Fig. 1d), which we confirmed was not due to cellular toxicity (Supplementary Fig. 1a). A similar reduction was also observed for coronaviruses (HCoV-229E and OC43) (Fig. 1e), but not for infection by influenza virus (which replicates in the nucleus) (Supplementary Fig. 1b). Virus production could be rescued in cells reconstituted with rUbe2g2[WT] but not rUbe2g2[C89K] suggesting that the catalytic activity of Ube2g2 is necessary for its role in the viral lifecycle (Fig. 1f). This phenomenon was also confirmed in hepatocytes, indicating that Ube2g2-dependent virus production was not a cell type specific effect (Supplementary Fig. 1c). We and others have previously shown that Ube2g2 interacts with Aup1 via its G2BR domain[8,16,17]. To test whether this interaction is retained in virus-infected cells, we immunoprecipitated endogenous Ube2g2 using protein G beads from mock and virus-infected cells. The eluate was resolved by SDS-PAGE and immunoblotted with antibodies for the indicated proteins in total cell lysates and IP elute. As anticipated, Ube2g2 interacted with Aup1 in both mock and virus-infected samples during the course of infection (Fig. 1g).

## Ube2g2 is necessary for virus replication

We next assessed the stage in the viral life cycle that requires Ube2g2. To exclude the possibility that Ube2g2 deletion disrupts viral entry, we took advantage of a fluorescence-based assay to evaluate entry and fusion of the virus with host endosomal membranes. Wild-type and Ube2g2[−/−] cells were infected with Zika virus pre-labelled with the fluorescent probe octadecyl rhodamine B (R18). R18 is self-quenched at high concentrations, but upon viral fusion with endosomal membranes, dilution of the R18-labelled virus leads to increased fluorescence intensity that can be quantified using flow cytometry. Viral internalisation can therefore be visualised by confocal imaging of R18 punctae (Fig. 2a). Red punctate structures indicating fusion of internalised virus particles with endosomal membranes were visible in both wild-type and Ube2g2[−/−] cells. We quantified the percentage of R18-positive cells as a read-out for the proportion of virus particles that gain entry into cells. Wild-type and Ube2g2[−/−] cells incubated with R18-prelabelled culture medium were used as negative control. ~40% of the population were R18 positive in both WT and Ube2g2[−/−] cells, with no significant difference detectable between the two, indicating that Ube2g2 has no impact on viral entry (Fig. 2b).

To evaluate the impact of Ube2g2 deletion on virus replication, we used a Zika replicon system carrying a Renilla luciferase-reporter. Zika replicon is a self-replicative viral RNA which expresses all the viral non-structural genes with the viral structural genes replaced with a luciferase reporter (Fig. 2c). Luciferase intensity is measured as a read out for viral replication. The replicon system therefore provides a useful tool to decouple viral replication from other confounding factors in the viral lifecycle such as entry, assembly and secretion. Zika replicons were transfected into wild-type and Ube2g2[−/−] cells; at indicated timepoints, the cells were lysed and the luciferase intensity measured (Fig. 2d). The luciferase intensity remained comparable at early time

points during translation, but starting from 6 h (at the replication phase) increased in the wild-type cells, while that in Ube2g2[−/−] cells remained significantly lower, indicative of defective replication (Fig. 2d). Accordingly, this defect in replication in Ube2g2[−/−] and rUbe2g2[C89K] cells was even more pronounced at later time points (24 and 48 h post transfection) (Fig. 2e, f). To confirm these data, we determined the intracellular Zika RNA production in infected wild-type and Ube2g2[−/−] cells using RT-qPCR. The results presented as fold difference between intracellular viral RNA in wild-type and Ube2g2[−/−] cells (2^-ΔΔCt (WT/Ube2g2[−/−])), indicate that viral genome replication was significantly lower in Ube2g2[−/−] cells compared to wild-type cells (Supplementary Fig. 2a).

To further confirm the defect in virus replication we visualised Zika virus replication sites using immunofluorescence. Wild-type cells infected with Zika for 24 h displayed a typical perinuclear distribution of viral E protein (Fig. 2g). Double-stranded RNA (dsRNA) generated as intermediates during virus RNA replication largely co-localised with Zika E protein in the wild-type cells. In contrast, virus-infected Ube2g2[−/−] cells displayed a dispersed E protein and dsRNA distribution, indicative of defective replication sites (Fig. 2g, h).

Furthermore, transmission electron microscopy (TEM) analysis of wild-type and Ube2g2[−/−] cells infected with Zika for 24 h revealed viral progenies in wild-type but significantly attenuated in the Ube2g2[−/−] cells. In addition, appearance of virus replication organelles could be seen in the wild-type cells, but were lacking in the Ube2g2[−/−] cells (Supplementary Fig. 2b–d). In Zika-infected wild-type cells, ER sheets were substantially dilated; large amount of virus-induced membrane invaginations, often referred to as vesicle packets were observed to accumulate. In comparison to WT, the typical condensed convoluted membranes and replication organelles were not observed in Ube2g2[−/−] cells. Instead, more autophagosome like double membrane vesicles and lysosomes were present in the Ube2g2[−/−] cells as described later (Supplementary Fig. 5).

## Defective expression of the viral replication complex in Ube2g2[−/−] cells

The Zika virus non-structural proteins interact with each other to form the replication complex detected as membrane invaginations formed upon ER-remodelling[3,20,21]. Furthermore, we previously reported that the viral NS4A-NS4B proteins were necessary for triggering lipophagy[8]. Given the defect in replication and replication organelle morphology (Fig. 2), we aimed to determine whether this complex is affected. We first measured the steady state expression levels of all viral proteins in the wild-type and Ube2g2[−/−] cells at 24 h and 48 h post infection (Fig. 3a). At both time-points, expression of the non-structural proteins was significantly attenuated in the Ube2g2[−/−] cells. On the other hand, expression of viral structural proteins prM and E were less affected at early time points (at 24 h post infection) but are significantly reduced at 48 h timepoint following multicycle replication. These data indicate that it is the replication complex that is specifically affected in Ube2g2-deficient cells. Loss in expression of the non-structural proteins e.g. NS5, NS4A-4B at earlier time points indicate that either synthesis or turnover of these proteins is impacted in Ube2g2[−/−] cells, resulting in defective replication and therefore a cumulative defect in viral protein expression at later timepoints (Fig. 3a).

To further investigate this phenotype and differentiate between defective synthesis versus increased degradation of the viral proteins, we measured the fate of newly synthesised viral proteins using pulse-chase in radiolabelled cells (Fig. 3b–h). Zika-infected wild-type, Ube2g2[−/−], rUbe2g2[WT] and rUbe2g2[C89K] cells (12 h, 24 h and 48 h post infection) were pulse labelled with [35S] cysteine/methionine and chased in cold media for the indicated time intervals. At each time point, viral proteins were immunoprecipitated, resolved by gel-electrophoresis and detected by autoradiography (Fig. 3b). In addition, production of virus

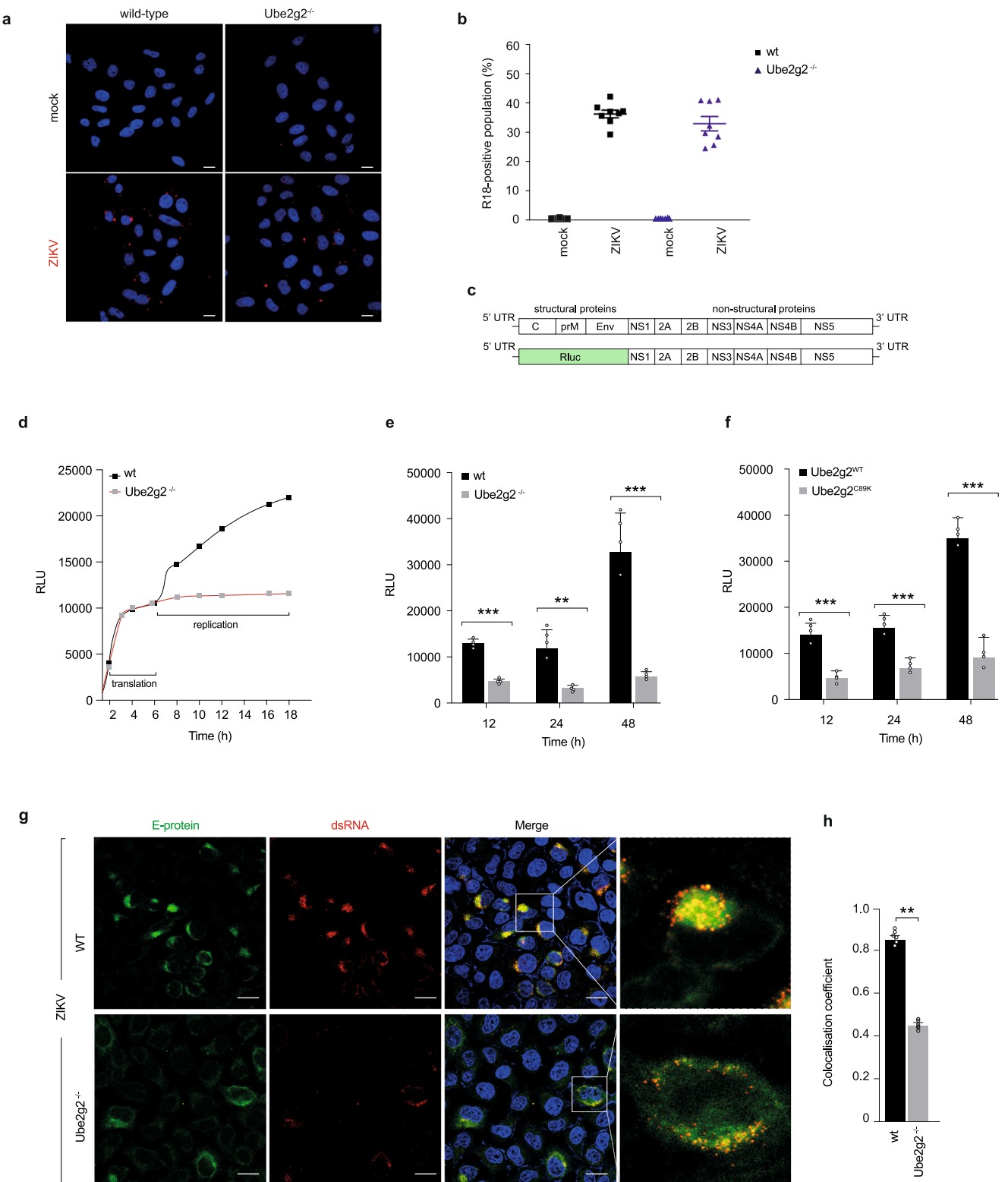

particles was also measured from supernatants of these cells (Fig. 3c). As observed with plaque assays, virus production from Ube2g2-deficient cells was substantially lower compared to Ube2g2-proficient cells (Fig. 3c). Interestingly, while synthesis of the structural proteins (E, prM) was only mildly slower in Ube2g2-deficient cells, particularly at 24 h post infection (Fig. 3d, g), synthesis of non-structural proteins was followed by their rapid degradation over time in the rUbe2g2$^{-/-}$ and rUbe2g2$^{C89K}$ cells (Fig. 3e–g). Quantitation of protein turnover therefore indicate

that the replication proteins do not form a stable complex in the absence of Ube2g2 (Fig. 3g), which very likely results in impaired membrane remodelling necessary for generating replication organelles. Interestingly, although synthesis of the structural proteins was not dramatically reduced in Ube2g2-deficient cells, they did not form the oligomeric complexes that is typical in wild-type cells (Fig. 3h). Collectively, these data indicate that defective membrane rearrangements underpin the block in biogenesis of virus replication organelles.

**Fig. 2 | Ube2g2-deficiency block virus replication. a** Wild-type and Ube2g2$^{-/-}$ cells were infected with either R18 labelled ZIKV at MOI 400 or pre-labelled control medium and visualised by immunofluorescence imaging. Hoechst staining indicates nuclei; red puncta structures indicate R18 labelled virus. **b** The R18 positive population was measured using flow cytometry at 2.5 hpi. Error bars represent mean ± SD, $n = 3$ biologically independent samples. **c** Schematic representation of Zika virus replicon; the structural gene segments are substituted with a luciferase reporter for activity readout. **d**–**f** Luminescence of ZIKV replicon RNA expressing luciferase in wild-type and Ube2g2$^{-/-}$ cells measured at indicated time points after transfection to measure translation (**d**) and virus replication in wild-type, Ube2g2$^{-/-}$, Ube2g2$^{WT}$ and Ube2g2$^{C89K}$ cells (**e**, **f**). Data are presented as mean ± SD, $n = 4$

biologically independent samples (**e**, **f**). Luciferase activity is shown as relative luminescence units (RLU). **$P < 0.01$; ***$P < 0.001$ (compared with wild-type by ANOVA followed by one-sided Dunnett's test). **g**, **h** Deletion of Ube2g2 affects the structure of intracellular viral replication and assembly complex. Wild-type and Ube2g2$^{-/-}$ cells were infected with Zika at MOI of 5. Cells were fixed at 24hpi and probed with anti-ZIKV envelope E-protein (488) and anti-double stranded (ds) RNA (555). Nuclei were visualised by Hoechst staining (**g**); Scale bar = 10 μM. Pearson's correlation coefficient was measured to determine co-localisation (**h**). Error bars represent mean ± SD, $n = 6$ biologically independent samples; **$P < 0.01$; ***$P < 0.001$ (compared with wild-type by ANOVA followed by one-sided Dunnett's test). (Source data are provided as a Source data file).

## Ube2g2 is required for virus-triggered lipophagy for assembly and secretion

A consistent feature of membrane remodelling during +RNA virus infection is selective autophagy of lipid droplets[3,7,9,10,22]. Our previous study demonstrated that Aup1 is important for virus-triggered lipophagy, for assembly and secretion of viral progenies[8]. Since Ube2g2 is a co-factor of Aup1 and was also identified in our initial screen, we aimed to determine whether it was also required for lipophagy and subsequent virus assembly/secretion. To specifically address assembly and secretion of virus particles, we took advantage of the Zika-prME virus like particles (Zika-VLP) system (Fig. 4a). These cells stably express Zika virus like particles containing only the virus structural proteins (prM and E) but not the viral nucleocapsid, thus allowing us to study the virus release in the absence of viral entry and replication[9]. We transfected cells constitutively expressing Zika-VLPs with either DsiRNA targeting Ube2g2 or non-targeting control DsiRNA, and the expression of intracellular (cell lysates) and extracellular (supernatant) VLP production was assessed using Western blotting (Fig. 4b).

Compared to Zika-VLP cells transfected with non-targeting siRNA, cells transfected with siRNA targeting Ube2g2 showed no detectable reduction in intracellular VLP production; however, a dramatic reduction of released mature form of VLPs was observed in the supernatants of Ube2g2-depleted cells (Fig. 4b, c). Lipophagy is critical for assembly and secretion of progeny virions[8,9]. Contribution of lipid droplet turnover at lysosomes was also reported for secretion of heavy cargo through the secretory pathway[23]. In line with these findings, Ube2g2-deficient cells displayed impaired secretion of VLPs as anticipated, in support of its role in lipophagy. In addition, the phenomenon of decreased lipid droplets in wild-type VLP-secreting cells, was inhibited in the Ube2g2$^{-/-}$ cells, also indicative of defective lipophagy (Fig. 4d, e). This defect in secretion was not specific for VLP secretion, but could also be observed as seen previously[23] for Vesicular stomatitis virus glycoprotein (VSV-G) (Fig. 4f, g). Wild-type and Ube2g2$^{-/-}$ cells expressing VSV-G were pulsed with [$^{35}$S]cysteine/methionine for 10 min and chased for indicated time intervals. Immunoprecipitated VSV-G were EndoH-treated to measure transport from the ER to the Golgi (Fig. 4f). VSV-G transport was quantitated as amount of EndoH resistant forms as a fraction of total (Fig. 4g). VSV-G transport was found to be impaired in the Ube2g2$^{-/-}$ cells compared to wild-type cells. In addition, the ER-resident pool of VSV-G was found to undergo degradation over time (Fig. 4f, g). Collectively, these data suggest that Ube2g2 might play a dual function in the viral life-cycle—first in the replication process via biogenesis of replication organelles, and second in lipophagy to facilitate secretion of assembled viral progenies.

To determine if lipophagy was affected in Ube2g2-deficient cells, we first stained LDs in cells expressing GFP-tagged Ube2g2 (Fig. 5a–d). Ube2g2 colocalised with lipid droplets, particularly in virus-infected cells, as observed in cells expressing Ube2g2-eGFP either pulsed with a fluorescent fatty acid reporter (Bodipy 558/568 C12) (Fig. 5a, b) or stained with Nile Red to visualise LDs (Fig. 5c, d). This was in concert with Aup1, which displayed a similar distribution in virus-infected cells (Supplementary Fig. 3a, b). Co-localisation of LDs could also be verified with lysosomal markers (Lamp1) in virus-infected cells, in particular

when treated with Bafilomycin A to impair lysosomal degradation (Supplementary Fig. 3c, d).

To measure lipophagy, we visualised LDs in wild-type and Ube2g2$^{-/-}$ cells at 48 and 72 h post infection. While in wild-type cells LDs underwent hydrolysis as anticipated, in Ube2g2$^{-/-}$ cells we observed an accumulation of LDs as measured by Nile red staining of LDs by immunofluorescence and flow cytometry (Fig. 5e, f). These data are in support of Ube2g2 facilitating virus-triggered lipophagy as part of the Aup1-Ube2g2 complex.

To determine how Ube2g2-deficiency impaired lipophagy, we first measured Aup1 levels in wild-type and Ube2g2-deficient cells. Defective lipophagy was not due to loss of Aup1, since its expression levels in Ube2g2-deficient cells remained unaffected even at late timepoints in infections (Fig. 6a). On the other hand, expression of Ube2g2 was reduced in Aup1$^{-/-}$ cells as previously reported[19]. This was even more pronounced in virus-infected Aup1$^{-/-}$ cells presumably due to increased ER turnover. We also measured induction of autophagy by immunoblotting for LC3. Interestingly, even in mock infection, basal levels of autophagosomes (LC-II) were significantly higher in the Ube2g2$^{-/-}$ cells, which was increased further in virus-infected cells (Fig. 6b). This effect was accompanied by an increase in expression of lysosomes as detected by Lamp2 (Fig. 6c). Confocal imaging of Aup1 and Lamp2 also revealed increased co-localisation of the two in Ube2g2$^{-/-}$ cells compared to that of wild-type (Supplementary Fig. 3a, b). To further characterise this process, we visualised autophagic flux in wild-type and Ube2g2$^{-/-}$ cells using cells stably expressing RFP-GFP-LC3. While increased autophagic flux was detected in wild-type cells upon virus infection, in Ube2g2$^{-/-}$ cells, high abundance of autolysosomes was detected even in uninfected cells, which remained at high levels upon infection (Fig. 6d, e), in line with the biochemical data. Quantification of autophagosomes versus autolysosomes by flow cytometry were also in line with confocal imaging analyses, showing increased autolysosomes in Ube2g2$^{-/-}$ cells (Fig. 6f). These data indicate that defective lipophagy is not on account of blocked autophagosome formation, but more likely due to aberrant autophagy in Ube2g2-deficient cells. To specifically rescue the phenotype of impaired lipophagy, we measured viral titres in cells exogenously supplied with free fatty acids[8,24]. Interestingly, while virus production in Aup1-deficient cells could be rescued by exogenous free fatty acids, this could not be rescued in Ube2g2-deficient cells (Fig. 6g), suggesting that Ube2g2 performs an additional function besides lipophagy in the viral life-cycle.

## Ube2g2-dependent degradation of stress chaperones is a key feature of membrane remodelling during biogenesis of virus replication organelles

Given the increased autophagosomal flux in parallel to impaired selective lipophagy, we hypothesised that the underlying mechanism of defective virus replication and assembly in Ube2g2-deficient cells was due to aberrant autophagy. Replication organelles are formed upon remodelling of the ER[5,25,26]. We therefore hypothesised that a combination of (i) defective lipophagy and (ii) increased ER-phagy, resulted in defective membrane remodelling required for biogenesis of replication organelles in flavivirus infection (Fig. 7a). The induction

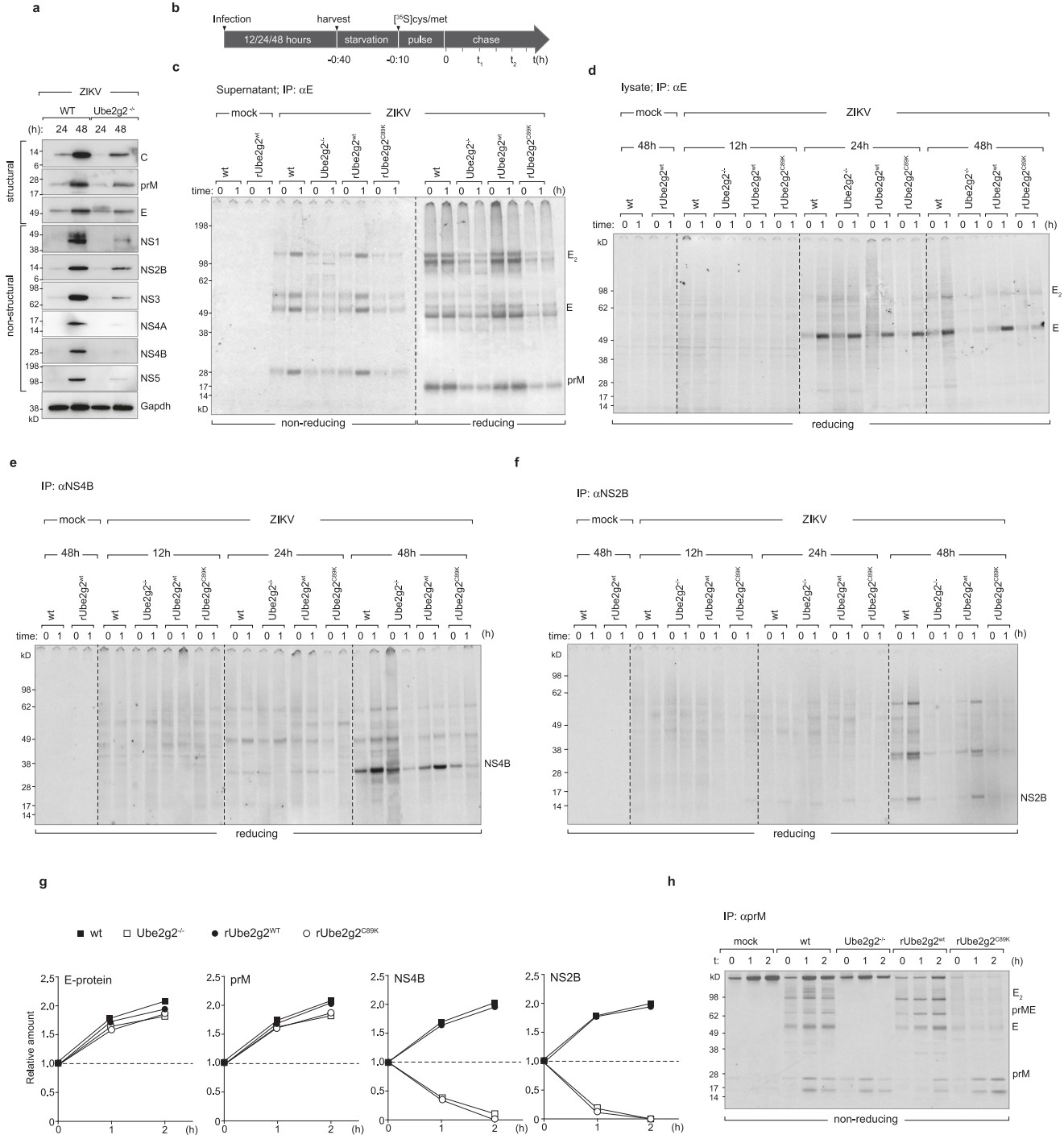

**Fig. 3 | Ube2g2-deficiency impairs biogenesis of virus replication organelles.**
**a** Wild-type and Ube2g2⁻/⁻ cells were infected with ZIKV at MOI = 2. The expression levels of structural and non-structural proteins were analysed by Western blotting. Data is representative of four biologically independent samples. **b** Schematic of pulse chase assay. Cells were harvested following infection for 12, 24 or 48 h. Cells were starved of cysteine/methionine for 30 min and pulsed with 20μCi of [³⁵S] cysteine/methionine for 10 min, followed by chase in cold media for indicated timepoints. Supernatants and cell pellets were collected at each time point for immunoprecipitation. **c** Released virus particles were collected from supernatants (from 48 h post infection) by immunoprecipitation on anti-E antibodies, subjected to non-reducing versus reducing (+βME) conditions, resolved by gel electrophoresis and detected by autoradiography. **d**–**f** Lysates from indicated cells were subjected to immunoprecipitation on anti-E, anti-NS4B and anti-NS2B for autoradiography. **g** Quantitation of indicated viral proteins from pulse-chase assays. Values are representative of at least four independent biological experiments. **h** Viral prM immunoprecipitated from radiolabelled cells under non-reducing conditions and detected by autoradiography. All images are representative of at least 3 independent biological replicates. (Source data are provided as a Source data file).

of UPR and ER-expansion is a necessary step for biogenesis of viral replication organelles[27–29]. Flaviviruses achieve this via a couple of means: they encode a cysteine protease NS3, which cleaves the ER-phagy receptor FAM134B, to induce ER expansion[30]. In addition, infection generates an oxidative environment beneficial to virus replication, which in turn results in stress to induce XBP1 splicing and induction of downstream effectors essential for viral replication complex formation[31–34].

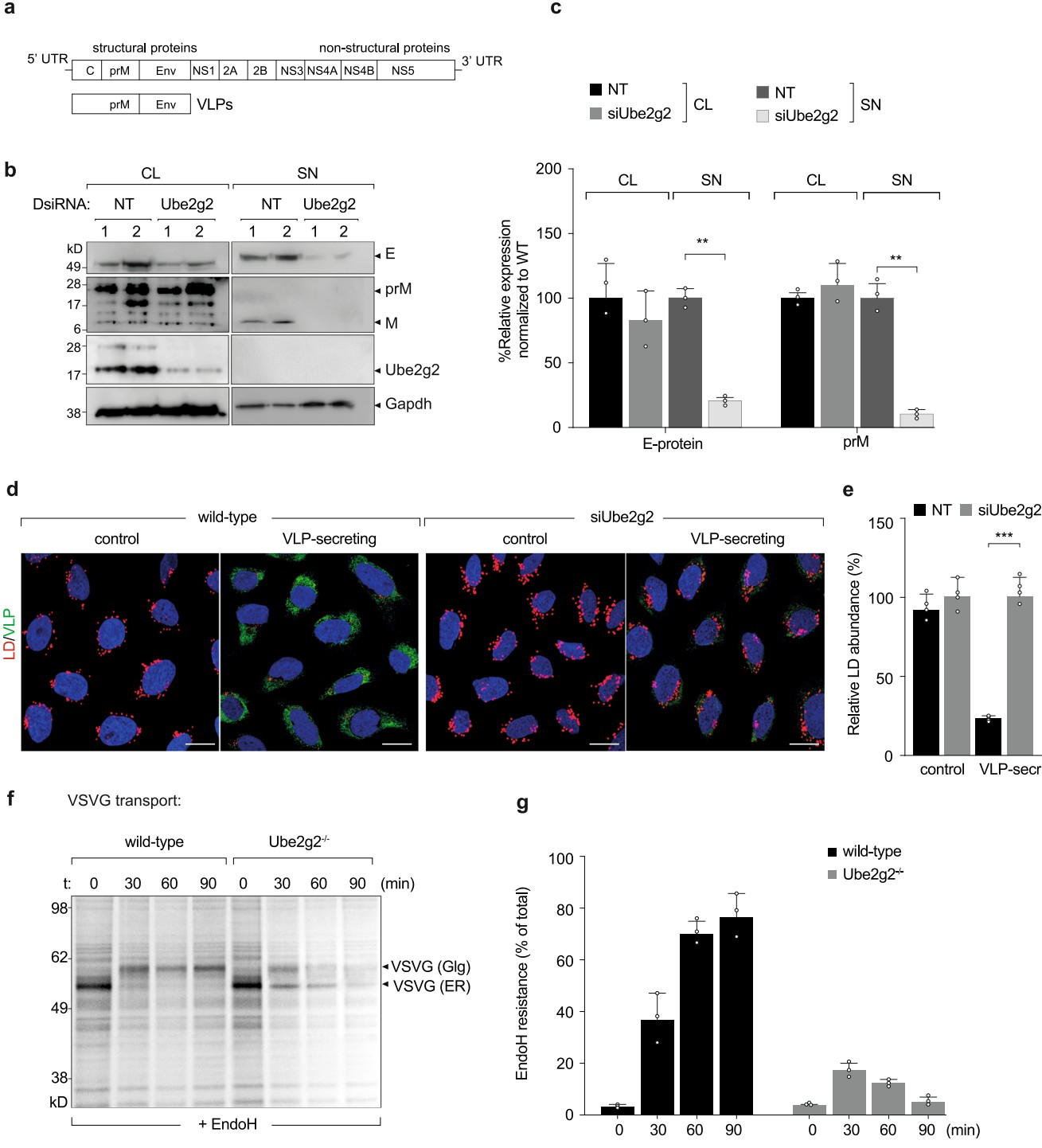

**Fig. 4 | Ube2g2 is required for virus secretion. a** Schematic of virus like particles. **b** HeLa cells constitutively secreting virus like particles (VLPs) were subjected to control and Ube2g2 targeting siRNA. Lysates and supernatants from cells were immunoblotted for viral prM and E proteins. **c** Quantitation of viral prM and E proteins in the lysates and supernatants of wild-type and Ube2g2-depleted cells. Error bars represent mean ± SD, $n = 3$ biologically independent samples; **$P < 0.01$; ***$P < 0.001$ (compared with wild-type by ANOVA followed by one-sided Dunnett's test). **d** VLP-secreting cells were treated with non-targeting and Ube2g2 targeting siRNA followed by Nile red staining for lipid droplets and anti-E staining for VLPs. **e** Abundance of lipid droplets was quantified by Nile red positive area over 500 cells. Error bars represent mean ± SD of relative LD abundance as % of control cells,

$n = 4$ biologically independent samples. **$P < 0.01$; ***$P < 0.001$ (compared with wild-type by ANOVA followed by one-sided Dunnett's test). **f** Wild-type and Ube2g2$^{-/-}$ cells expressing vesicular stomatitis virus glycoprotein were pulsed with [$^{35}$S]cysteine/methionine and chased in cold media for indicated time intervals. VSV-G was immunoprecipitated at each time point and treated with EndoH at 37 °C for 1 h, resolved by SDS-PAGE and detected by autoradiography. Image is representative of 3 independent experiments. **g** Quantitation of EndoH-resistant forms of VSVG was determined from densitometric analyses of autoradiograms. Data is presented as mean ± SD, $n = 3$ biologically independent samples. (Source data are provided as a Source data file).

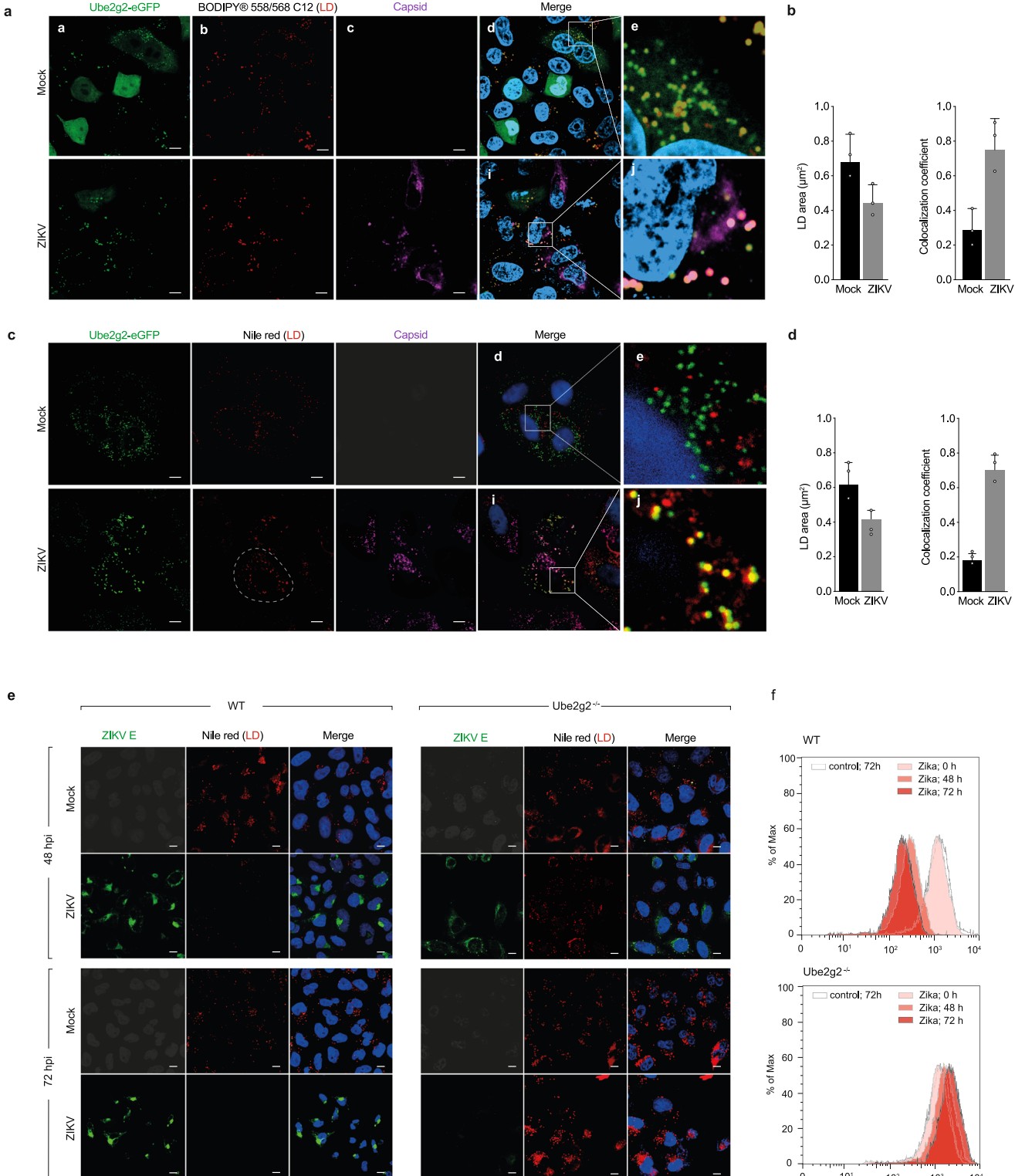

**Fig. 5 | Ube2g2 regulates lipophagy in virus-infected cells. a–d** Wild-type cells expressing GFP-Ube2g2 were either mock or Zika-infected at a MOI of 5. Cells were fixed at 24hpi and probed for Ube2g2 and LDs. LDs were stained with either BODIPY-C12 (**a**, **b**) or Nile red dye (**c**, **d**). Nuclei were stained with Hoechst; (Scale bars = 5 μm). Co-localisation of Ube2g2 with LDs was measured using Pearson's correlation coefficient. Quantification of the average Nile red-positive area per cell (μm²) was performed using "Analyze Particles" macro in ImageJ from >500 cells per condition; outlined region demarcates area per cell. Data is presented as mean ± SD, $n = 3$ biologically independent samples (**b**, **d**). **e** Wild-type and Ube2g2$^{-/-}$ cells were infected with ZIKV at MOI 5. Cells were fixed at 48hpi and 72hpi and probed with anti-ZIKV envelope protein (488). LDs were stained with Nile red dye. Nuclei were stained with Hoechst. (Scale bars = 5 μm). **f** LDs in mock and Zika-infected wild-type and Ube2g2$^{-/-}$ cells were quantitated by flow cytometry at 48 h and 72 h post infection. (Source data are provided as a Source data file).

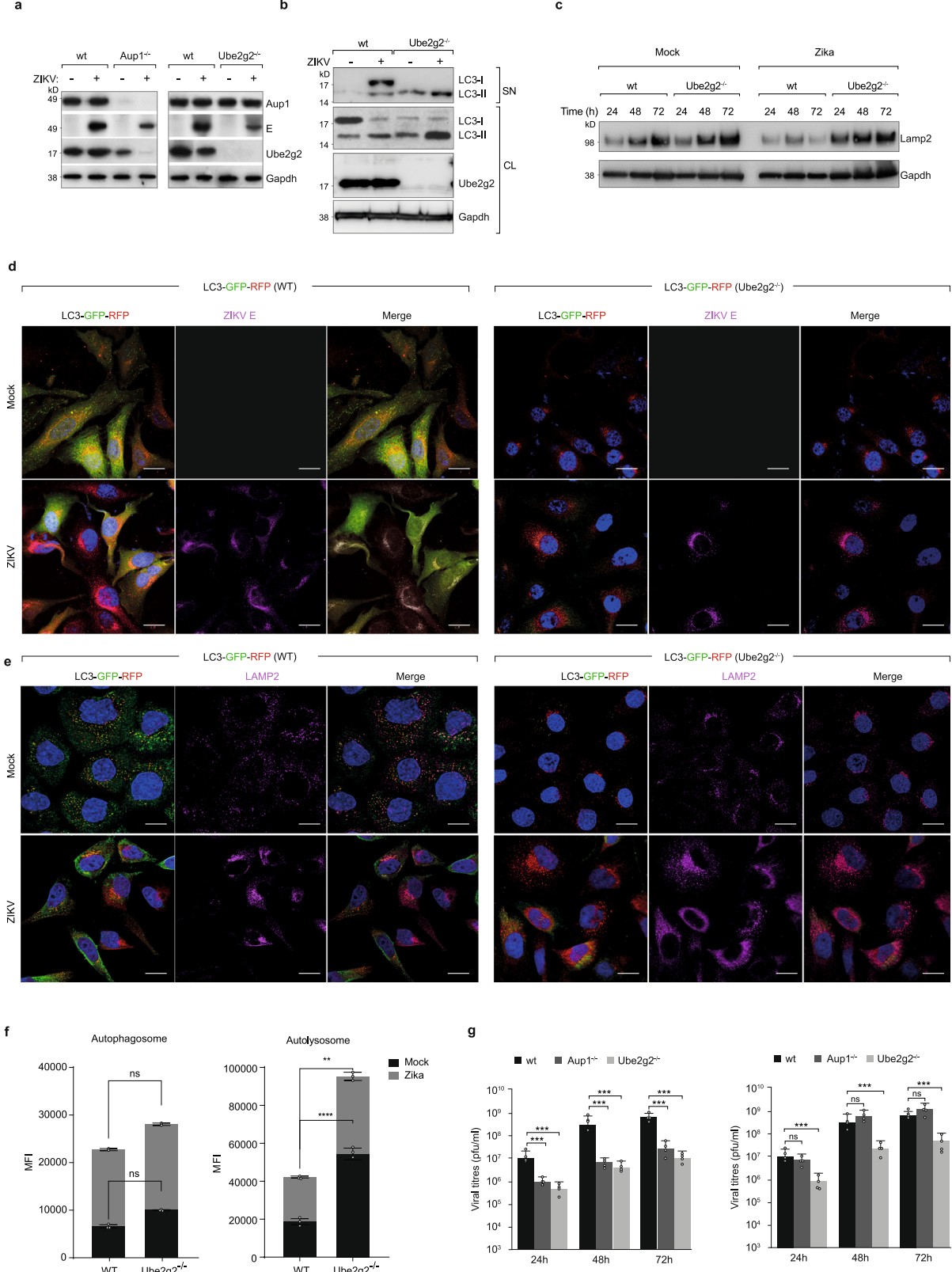

To verify UPR induction, we measured XBP1 splicing in virus-infected cells. In both wild-type and Ube2g2-deficient cells, infection was accompanied by XBP1 splicing (Fig. 7b, c). Spliced XBP1 was detected by RT-qPCR (Fig. 7b, c) as fold change compared to wild-type mock-infected cells. While in both the wild-type and Ube2g2-deficient cells, virus infection triggered increased Xbp1 splicing, the basal levels

of spliced Xbp1 were substantially higher in Ube2g2[−/−] cells compared to that of wild-type (Fig. 7b, c). UPR induction was therefore comparable in wild-type and Ube2g2-deficient cells. One of the viral strategies of inducing ER-expansion is via cleavage of the ER-phagy receptor FAM134B[30,35,36]. To evaluate virus-triggered ER expansion, we measured FAM134B expression in wild-type and Ube2g2-deficient cells

**Fig. 6 | Ube2g2-deficiency results in excessive autophagic flux. a**. Lysates from mock or Zika-infected wild-type, Aup1$^{-/-}$ and Ube2g2$^{-/-}$ cells (48 hpi) were immunoblotted for viral E-protein along with Aup1 and Ube2g2. **b**, **c** Supernatants and lysates from wild-type and Ube2g2$^{-/-}$ were immunoblotted for LC3 (**b**) and Lamp2 (**c**). **d** Wild-type and Ube2g2$^{-/-}$ cells stably expressing LC3-GFP-RFP were infected with ZIKV at MOI of 5. Cells were fixed at 24hpi and probed with anti-E protein (magenta) to visualise autophagosomes/lysosomes and viral protein distribution. Hoechst staining (Blue) was used to stain nuclei. (Scale bars = 10 μm). **e** Wild-type and Ube2g2$^{-/-}$ cells stably expressing LC3-GFP-RFP (HeLa-Difluo) were infected with ZIKV at MOI = 5. Cells were fixed at 24hpi and probed with anti-LAMP2 (647) to visualise autophagosomes/autolysosomes and lysosomes. (Scale bars = 10 μm).

Hoechst staining (Blue) was used to stain nuclei. **f** Abundance of autophagosomes and autolysosomes was determined using flow-cytometry using cells described above (**d**, **e**). Data are presented as mean ± SD, $n = 3$ biologically independent samples; **$P < 0.01$; ****$P < 0.0001$ (compared with wild-type by ANOVA followed by one-sided Dunnett's test). **g** Wild-type, Aup1$^{-/-}$ and Ube2g2$^{-/-}$ cells were mock or virus-infected and grown in medium supplemented with -/+ FFAs (10 mM) conjugated to BSA for indicated time points. Viral titres were measured using plaque assay. Data are presented as mean ± SD, $n = 4$ biologically independent samples; ***$P < 0.001$ (compared with wild-type by ANOVA followed by one-sided Dunnett's test). (Source data are provided as a Source data file).

(Fig. 7d). As anticipated, FAM134B cleavage remained unaltered between virus-infected Ube2g2 proficient and deficient cells across all time points, verified by siRNA-mediated depletion (Supplementary Fig. 6a, b). The abundance of FAM134B however was found to decrease over the time course of infection in Ube2g2-deficient cells, most likely due to FAM134B-independent ER-phagy (Fig. 7d).

ER expansion is typically followed by stress recovery of the ER to prevent apoptosis and occurs via degradation of specific chaperones. In particular, Herp/Herpud1, Sel1L, Hsp70 are rapidly downregulated to initiate recovery[37-39]. To measure rescue and remodelling of the ER we therefore measured specific chaperones associated with ER rescue (Fig. 7e). Infected wild-type and Ube2g2-deficient cells were pulsed with [$^{35}$S]cysteine/methionine and chased for 45 mins. Interestingly, in infected wild-type cells, Herpud1, Sel1L and calreticulin underwent rapid degradation following infection in wild-type and rUbe2g2$^{WT}$ cells, which was blocked in both the Ube2g2$^{-/-}$ and rUbe2g2$^{C89K}$ cells with increase in synthesis over time. Synthesis and turnover of Hsp70 on the other hand remained unaffected (Supplementary Fig. 5c). Other ER-phagy receptors including Rtn3[40,41], Fam134a, Fam134c remained unaffected. Atl3, another ER-phagy receptor also shown to be important for ER remodelling for dengue/Zika production[42] was degraded in Ube2g2-deficient cells (Fig. 7e and Supplementary Fig. 6c). In addition, expression of Sec62, another ER-phagy receptor[43-46], was induced specifically in the Ube2g2-deficient cells. Our data therefore indicate that Ube2g2 deficiency resulted in stabilisation of stress response chaperones along with increased expression of Sec62, which consequently triggers Sec62-dependent ER-phagy during infection.

Sec62-mediated ER-phagy was recently described as Chmp4 dependent[46,47]. To test if this pathway was triggered in Ube2g2-deficiency, we depleted either Sec62 or Chmp4 in the wild-type, Ube2g2$^{-/-}$, rUbe2g2$^{WT}$ and rUbe2g2$^{C89K}$ cells (Fig. 7f). These cells were grown in media supplemented with free fatty acids to circumvent the lipophagy defect as shown in Fig. 6, and challenged with Zika to measure production of infectious virus particles (Fig. 7g, h). Interestingly, in both Sec62 and Chmp4-depleted cells, defective virus production upon Ube2g2-deficiency was rescued to wild-type levels, in support of the hypothesis that Ube2g2 inhibits this pathway during virus infection (Fig. 7g, h). To determine whether the ER-stress chaperones were driven toward rapid degradation by the viral protease, we expressed NS2B-NS3 alone in wild-type, Ube2g2$^{-/-}$, rUbe2g2$^{WT}$ and rUbe2g2$^{C89K}$ cells (Fig. 7i). The rapid turnover of the stress response proteins was recapitulated in the wild-type and rUbe2g2$^{WT}$ cells, confirming that the viral protease was indeed able to downregulate them (Fig. 7i). Moreover, this effect was blocked in the Ube2g2-deficient cells, verifying that degradation was due to Ube2g2-mediated ubiquitylation (Fig. 7i and Supplementary Fig. 6d). Altered morphology of the ER with increased abundance of lysosomes was also observed by TEM analyses (Supplementary Fig. 5a, b) and by co-staining Lamp1 and Sec62 in Ube2g2$^{-/-}$ cells (Supplementary Fig. 5d). Collectively, these data highlight that selective autophagy plays a crucial role in the viral life-cycle. While lipophagy is essential for assembly and secretion of viral progenies, inhibition of ER-phagy is equally critical to biogenesis of viral replication organelles.

## Discussion

Infection of mammalian cells with +RNA viruses provides a unique model for investigating ER membrane dynamics. This is on account of the large malleable ER membrane that viruses use as a scaffold to form replication organelles[48]. In line with this, infection by flaviviruses such as dengue, Zika and coronaviruses[49,50] triggers ER expansion followed by extensive membrane remodelling to form virus replication compartments along ER-derived membranes. This model is supported by experimental data showing that virus infection is accompanied by activation of the unfolded protein response as previously reported[28], which we speculate enables membrane expansion and proliferation. Previous studies also reported that Zika/dengue NS3 protease is able to cleave the ER-phagy receptor (FAM134B), which would prevent ER turnover and result in membrane expansion[30]. In addition, we previously reported that lipid droplets are hydrolysed following infection to provide membrane components for biogenesis and secretion of viral progenies along the ER[8]. However, the sequence of events starting with induction of the UPR, followed by ER expansion, recovery and lipophagy that ultimately results in ER remodelling are not understood. In this study we identified Ube2g2 as a critical factor that enables ER remodelling to form the replication organelles necessary for virus replication and assembly.

Infection by flaviviruses places a massive burden on the ER to synthesise viral and host proteins necessary for replication, resulting in ER stress. The UPR activates several branches to enhance protein folding. In particular, the IRE1 pathway is triggered to generate the spliced version of XBP1, resulting in synthesis of chaperones and ERAD components. The question of how virus infection resolves this ER stress to prevent apoptosis before replication and production of progenies is accomplished is an unexplored one. This study provides mechanistic insights into virus-triggered downregulation of a subset of ER stress response proteins to enable remodelling of the membrane and form replication organelles, which is driven by the Ube2g2 ubiquitin conjugating enzyme.

Deficiency of Ube2g2 resulted in a combined defect of lipophagy and ER reorganisation, resulting in attenuated replication and degradation of the viral replication complex. By delineating specific steps in ER expansion and remodelling, we determined that Ube2g2 was required for Aup1-Ube2g2 dependent lipophagy (necessary for secretion of virus progenies) as well as for degrading chaperones to prevent apoptosis and maintain a state of membrane proliferation. These two functions of Ube2g2 are separable, both requiring the catalytic activity of Ube2g2. Loss of Ube2g2 activity resulted in stabilisation of the ER chaperones, which in turn triggered ER-phagy specifically via the Sec62-Chmp4 pathway, degrading the replication complex, thereby inhibiting virus replication. This study therefore provides mechanistic insights into the specific intermediate steps and the key role of Ube2g2 in the process of ER remodelling for biogenesis of viral replication organelles. Furthermore, it underscores the exquisite balance that is maintained in selective autophagy of specific organelles that determine outcomes in the viral lifecycle.

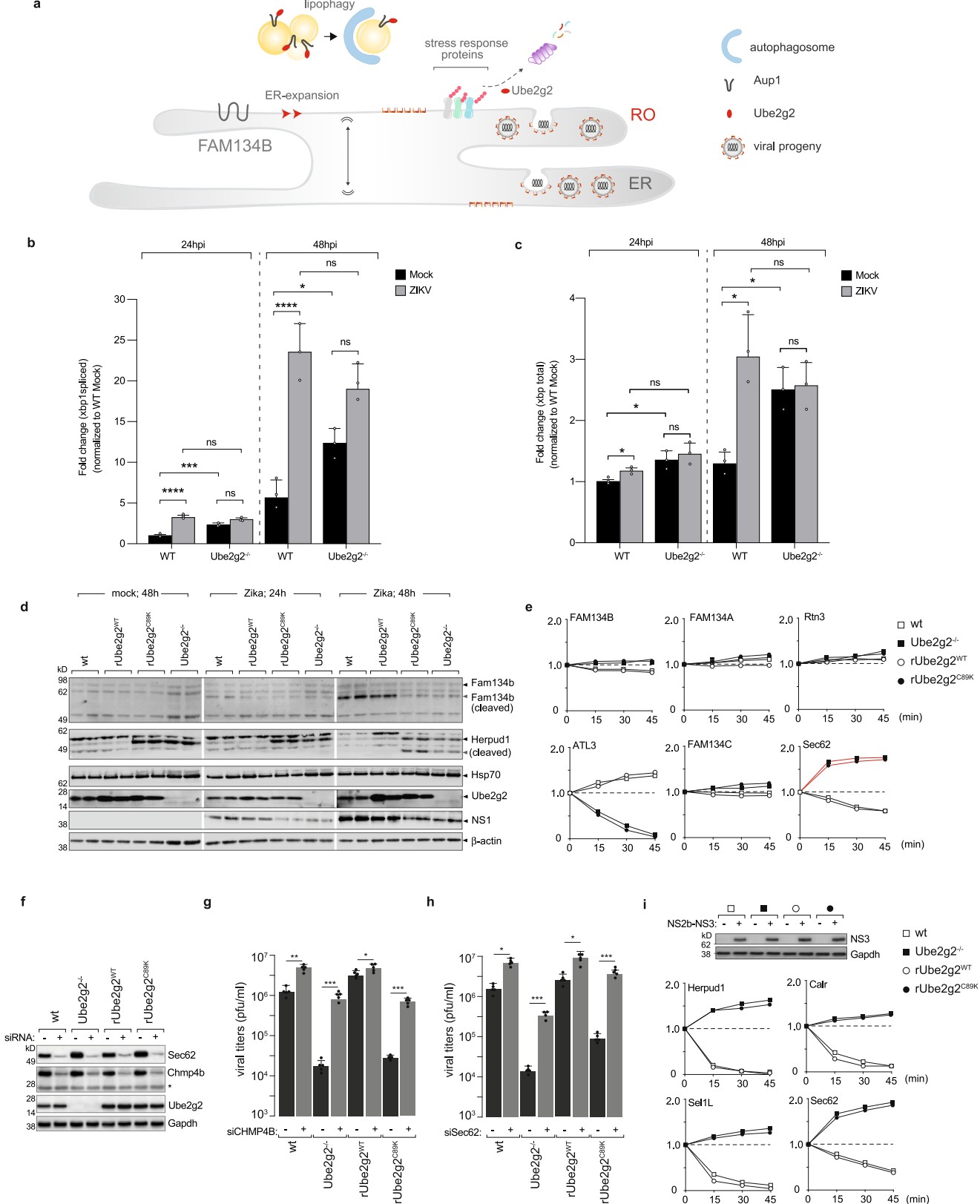

## Methods

### Generation of cell lines

Knockout and reconstituted versions of HeLa cells were generated for Ube2g2, namely, HeLa Ube2g2$^{-/-}$ and Hela-Difluo™ hLC3-Ube2g2$^{-/-}$ and HeLa rUbe2g2 using sgRNA sequences cited in Table 1. In-house generated HeLa-ZIKV-PrME virus like particle producing cell line was

depleted in Ube2g2. Cells were maintained in DMEM supplemented with 10% FBS and 1% penicillin and streptomycin.

Potential target sequences for CRISPR interference were found using published protocols. The knockout cells were generated using sgRNAs designed to target the protein of interest and cloned into the CRISPR/Cas9 vector pX459 (Table 1). The pX459-sgRNA clone was

**Fig. 7 | Ube2g2 triggers rapid turnover of stress response genes to prevent ER-phagy. a** Schematic of the role of Ube2g2 in triggering selective lipophagy and inhibiting ER-phagy. **b, c** Wild-type and Ube2g2$^{-/-}$ cells were mock or virus-infected at MOI 2 for indicated time points. At each time point, Xbp1 splicing was measured by qPCR. Data are presented as mean ± SD, $n = 3$ biologically independent samples; $*P < 0.05$; $***P < 0.001$; $****P < 0.0001$ (derived from unpaired Student's $t$ test for within group comparisons). **d** Expression of FAM134B, Herpud1, Hsp70 was measured by immunoblotting in wild-type, Ube2g2$^{-/-}$, rUbe2g2$^{WT}$ and rUbe2g2$^{C89K}$ cells at indicated timepoints post infection. **e** Turnover of newly synthesised ER proteins indicated was quantitated using pulse chase in [$^{35}$S]cysteine/methionine labelled cells. **f** Immunoblots were performed in lysates from indicated cells (wild-type,

Ube2g2$^{-/-}$, rUbe2g2$^{WT}$, rUbe2g2$^{C89K}$) expressing non-targeting siRNA as control or against Chmp4 or Sec62. Gapdh was measured as loading control. **g, h** Viral titres were measured in cells described in (f) grown in media supplemented with 10 mM FFA conjugated to BSA, using plaque assays. Data represent mean ± SD $n = 5$ biologically independent samples; $*P < 0.05$; $**P < 0.01$; $***P < 0.001$; $****P < 0.0001$ (compared with wild-type by ANOVA followed by one-sided Dunnett's test). **i** Zika NS2b-NS3 was expressed in wild-type, Ube2g2$^{-/-}$, rUbe2g2$^{WT}$, rUbe2g2$^{C89K}$; empty vector was used as control. Expression was detected using immunoblotting. Turnover of indicated ER proteins was quantitated by pulse chase in [$^{35}$S]cysteine/methionine labelled cells. (Source data are provided as a Source data file).

---

transfected into HeLa cells using FuGENE HD Transfection reagent (Promega, USA). After 24 h, cells were subjected to puromycin selection at a concentration of 1.5 to 3 μg/ml for 3 days, followed by limiting dilution to isolate single colonies, which were expanded and verified for deletion by immunoblotting.

Reconstituted Ube2g2 HeLa cells were generated by subcloning the lentiviral vector pcHMWS-IRES-hygromycin to carry Ube2g2 or that mutated by site-directed mutagenesis to C89K-Ube2g2 cDNA carrying a hygromycin resistance cassette. The four constructs were transfected into 293T cells with pHKIP115 (HIV gag, pol) and pHKIP131 (VSV-G) using PEI as transfection agent. Pseudoviruses were collected and stably transduced into the HeLa Ube2g2$^{-/-}$ cells. The cells underwent hygromycin selection (500 μg/mL) and single clones were selected and evaluated using immunoblotting for Ube2g2. These clones were included in infection assays alongside wt and Ube2g2$^{-/-}$ to evaluate viral production. HeLa Ube2g2-eGFP-Aup1-mCherry cells were similarly generated by a two-step selection procedure with pseudoviruses generated with the lentiviral pcHMWS-IRES-hygromycin to carry Ube2g2-eGFP and Aup1-mCherry respectively. HeLa Ube2g2-eGFP cells were generated first as described above and underwent hygromycin selection. HeLa Ube2g2-eGFP-Aup1-mCherry were then generated based on HeLa Ube2g2-eGFP.

### Virus stocks
ZIKV (NC-14-5132) and DENV2 (16681) were propagated in the Mosquito *Aedes albopictus* C6/36 cell line and titrated using plaque assay in Vero E6 cells, challenged with 10-fold serial dilutions of virus for 1 h at 37 °C. Cells were subsequently incubated in an overlay of methyl-cellulose-based medium (MEM 2.85 g, 1.2% (w/v) Methyl Cellulose, 1% 1 M HEPES, 1.5% 0.2 M Glutamine, 3% (v/v) 7% NaHCO₃, 10% FBS). Virus titres were determined after 6 days of incubation using crystal violet (0.1% (w/v) crystal violet, 20% EtOH, 80% dH₂O) to visualise the plaques.

HCoV-229E and OC43 were a kind gift from Dr Chris Mok. Viral titres were determined with standard TCID50 assay in MRC5 cells. Cells

### Table 1 | Oligonucleotides used in this study

| | | |
|---|---|---|
| Primers | ZIKV envelope primers | **Fwd:** 5′-TTGGGTTGTGTACGGAACCTG-3′ **Rev:** 5′-GTGCTTTGTGTATTCTCTTGA-3′ |
| | GAPDH primers | **Fwd:** 5′-GGAGCGAGATCCCTCCAAAAT-3′ **Rev:** 5′-GGCTGTTGTCATACTTCTCATGG-3′ |
| | primer (cloning) | **Fwd** (no-FLAG): 5′-ATATGGATCCGCCACC ATGGCGGGGACCGCGCTCAAG-3′ **Rev:** 5′-CACGCTCGAGTCACAGTCCCAGA-GACTTCT GGACGATC-3′ |
| | Spliced Xbp1 (set 1) | **Fwd:** 5′-TGC TGA GTC CGC AGC AGG TG-3′ **Rev:** 5′-GCT GGC AGG CTC TGG GGA AG-3′ |
| | Spliced Xbp1 (set 2) | **Fwd:** 5′-GCT GAG TCC GCA GCA GGT-3′ **Rev:** 5′-CTG GGT CCA AGT TGT CCA GAA T-3′ |
| | Total Xbp1 | **Fwd:** 5′- TGA AAA ACA GAG TAG CAG CTC AGA –3′ **Rev:** 5′-CTG GGT CCA AGT TGT CCA GAA T-3′ |
| sgRNA | ube2g2_A1_s | CACCG-GTTGGGATGAAACATCTCAC |
| | ube2g2_A1_a | AAACGTGAGATGTTTCATCCCAACC |

were seeded in 96-well plates and incubated at 37 °C, 5% CO₂ until 80–90% confluent and challenged with half-log serial dilutions of virus for 3–4 days. Cells were then inspected for the presence or absence of cytopathic effect, and the TCID50/mL was calculated.

### Antibodies, plasmids and oligonucleotides
Rabbit α-Aup1 Ab101984 (Abcam, UK; 1:1000); Rabbit α-Ube2g2 Ab174296 (Abcam, UK; 1:1000); Rabbit α-Perilipin Ab172907 (Abcam, UK; 1:1000); Rabbit α-Hrd1 LS-C668919 (LSBio, WA, USA; 1:1000); Rabbit α-Gp78 ab101284 (Abcam, UK; 1:1000); Rabbit α-Herpud1 26730 (Cell Signalling Technology, MA, USA; 1:500); Rabbit α-LC3b 2775S (Cell Signalling Technology, MA, USA; 1:1000); α-Rabbit LAMP1 (D2D11) 9091S (Cell Signalling Technology, MA, USA; 1:1000); Mouse recombinant α-(linkage specific K63)-ubiquitin Ab179434 (Abcam, UK; 1:1000); Mouse recombinant α-(linkage specific-K48) ubiquitin Ab140601 (Abcam, UK; 1:1000); Mouse α-Ubiquitin Ab7780 (Abcam, UK; 1:1000); Rabbit polyclonal α-FAM134C Ab202125 (Abcam, UK; 1:1000); α-Reticulon-3 (A302-860A, Cambridge Bioscience Ltd; 1:1000); α-FAM134B (61011), (Cat no 61011S, Cell Signaling Technology; 1:1000); HSP70 (10995-1-AP, Proteintech Europe Ltd; 1:1000); Mouse α−4G2 In-house produced (1:5000); Mouse α-GAPDH; Goat α-mouse Alexafluor®488 Ab150117 (Abcam, UK; 1:10,000); Goat α-rabbit Alexafluor®488 Ab150077 (Abcam, UK; 1:10,000); Goat α-mouse Alexafluor®647 Ab150119 (Abcam, UK; 1:10,000); Goat α-rabbit HRP 7074 (Cell Signalling Technology, MA, USA; 1:10,000); Goat α-mouse HRP Ab97040 (Abcam, UK; 1:10,000); Goat α-Rabbit Texas Red® Ab6719 (Abcam, UK; 1:10,000); Rabbit α-prM GTX133305 (GeneTex, USA; 1:5000); Rabbit α-NS3 GTX133309 (GeneTex, USA; 1:1000); Rabbit α-NS4A GTX133704 (GeneTex, USA; 1:5000); Rabbit α-NS4B GTX133311 (GeneTex, USA; 1:1000); Rabbit α-NS1 GTX133307 (GeneTex, USA; 1:1000); Rabbit α-NS5 GTX638132 (GeneTex, USA; 1:1000); Rabbit α-NS2B GTX GTX133308 (GeneTex, USA; 1:500); Rabbit α-Capsid GTX GTX133317 (GeneTex, USA; 1:1000)

### Plasmids
pX459-U6-Chimeric_BB-CBh-hSpCas9 (Addgene #42230), pcHMWS-Ube2g2-IRES-Hygromycin (cloned in this study), pcHMWS-Ube2g2-Flag-IRES-Hygromycin (cloned in this study), pHKIP115 (HIV gag-pol), pHKIP131 (VSV-G).

### Viral production
HeLa wt, Ube2g2$^{-/-}$, rUbe2g2$^{WT}$ and rUbe2g2$^{C89K}$ cells were plated in 6- or 12-well plates and infected at MOI of 2. At the time points indicated in the figures, the supernatants and cell lysates were collected. The cell lysates were collected in 200 μL RIPA-buffer (1% (v/v) Triton X-100, 50 mM Tris-HCl, pH=7.4, 150 mM NaCl, 1 mM EDTA, 0.5% (w/v) sodium deoxycholate). The supernatants were used for quantitation of viral production by plaque assay and RT-qPCR. Supernatants were serially diluted to quantify viral production by means of infectious particles in plaque assays. The samples were 10-fold serially diluted and added to a Vero cell monolayer. After a 60 min adsorption at 37 °C, 5% CO₂, cells were washed with PBS, and overlaid with methyl cellulose-based media (10% FBS). After 6 days of incubation at 37 °C, 5% CO₂ the cell

monolayers were fixed and stained with crystal violet for quantification of the plaque forming units. Results were presented as means ± standard deviation (SD), unless stated otherwise.

The supernatants were simultaneously used to assess extracellular viral RNA levels, by extracting total RNA using the TaKaRa viral extraction kit (Takara Bio Inc., Japan). Real-time PCRs were performed using one-step SYBR green kit (Takara Bio Inc., Japan) with E-protein-specific primers (Table 1). A standard curve was included of 10-fold serial dilutions of ZIKV RNA with confirmed plaque-forming units/mL (pfu/mL). Hereby, the recorded Cp value of the RNA samples was translated into pfu/mL values. All samples were measured in technical triplicates. Results obtained from RT-qPCR were presented as means ± standard deviation (SD), unless stated otherwise. For coronaviruses, wild-type and Ube2g2$^{-/-}$ were seeded in 24-well plate and infected with HCoV-229E and OC43 at MOI 0.1. Twenty-four hours post infection, supernatants were collected and quantified for viral production by TCID50 assay as described above in MRC5 cells.

### R18-fusion assay

To determine whether Ube2g2 deletion can disrupt viral entry specifically, a membrane fusion assay was set up to evaluate viral fusion in wild-type and Ube2g2$^{-/-}$ cells. Cells were infected with ZIKV labelled with the fluorescent probe octadecyl rhodamine B Chloride (R18). R18 is self-quenching at high concentrations and shows no fluorescence signal when prelabelled on virus. However, as R18-labelled virus fuses with unlabelled cellular membranes, R18-dilution occurs leading to increased fluorescence intensity. This increased fluorescence signal was quantified as a measure of viral fusion using flow cytometry and confocal microscopy.

Pre-labelling of ZIKV with R18 was executed by incubation of ZIKV or control media with 1:1000 2× R18 dye (Thermo Fisher) for 1 h on ice. Residual dye was removed from the labelled virus by ultracentrifugation on a 25% sucrose gradient at 28000 rpm for 1 h. After resuspension of the virus on ice, cells were incubated with the virus at MOI 100 (as determined prior to centrifugation) for 1 h at 4 °C followed by 1 h at 37°C. Afterwards the cells were washed, trypsinised and fixed (4% PFA) for analysis on flow cytometry (Attune NxT, Invitrogen). The %-R18+ population was gated using SSC, FSC and the YL1 Emission Filter (585/16 nm) and was analysed using FlowJo (OSX64-10.5.3). To confirm results found by flow cytometry, internalisation of R18-labelled virus was also visualised using confocal microscopy. In this set-up, the cells were infected with R18-labelled ZIKV at MOI 400 for 1 h at 4 °C and fixed after 0.5, 1,5- and 2.5-h incubation at 37 °C post infection. The fixed cells were stained with Hoechst staining (Thermo Fisher) for 15 min at 37 °C and observed under the LSM confocal microscope.

### Luciferase reporter-based replicon assay

A Luciferase reporter based ZIKV replicon assay was set up to assess viral replication (the replicon plasmids were kindly provided by Dr Chengfeng Qin (Beijing Institute of Microbiology and Epidemiology)). ZIKV Renilla luciferase replicons were transcribed using a Ribomax T7 RNA polymerase kit (Promega, USA). The resulting RNA was purified using TaKaRa MiniBEST Universal RNA Extraction Kit (Takara Bio Inc., Japan). The wild-type and Ube2g2$^{-/-}$ cells were seeded in 24-wells plates 24 h prior to transfection, washed with PBS and transfected using Lipofectamine 3000 (Thermo Fisher, USA) as transfection agent. Lysates were collected at indicated time-points in lysis buffer diluted with luciferase assay buffer (1:4 ratio). Luciferase expression was measured using the Renilla Luciferase Assay system (Promega, USA).

### VLP release assay

The in-house generated HeLa-ZIKV-PrME cell line was used to assess maturation and release of the viral particles. This cell line stably expresses (and secretes) ZIKV virus-like particles consisting of the structural proteins Pr, M and E. In this assay, the cells were plated in duplicate at 2×10$^6$ on 10-cm dishes. Cells were transfected with either DsiRNA targeting Ube2g2 or scrambled non-targeting DsiRNA (Tri-FECTa® RNAi Kit, Integrated DNA Technologies, USA). DharmaFECT™ (Horizon Discovery, UK) was used for transfections, as per manufacturer's protocol. At 48 hp and 72 hp post infection, complete media supplemented with FBS was changed for Gibco OptiMEM™ (Thermo Fisher) and cells were incubated with OptiMEM for a further 24 h to allow VLPs to release into the supernatant. At 48 and 72 h post transfection, the supernatants and cell lysates were collected. Supernatants were concentrated by Amicon® Ultra-15 centrifugal filter units at 4 °C at 4000 × g for 50 min. The lysates were collected in 500 μL RIPA-buffer. Cellular debris was removed from the lysate by freeze-thawing the samples three times and spun down at 18,624 × g at 4 °C. Protein concentrations of the cell lysates and supernatants were measured on Nanodrop™ (Thermo Fisher, MA, USA) and used for preparation of the cell lysate and supernatant samples for immunoblotting. Western blot quantifications were performed using ImageJ.

### Western blotting

For immunoblotting, lysates collected in RIPA-buffer were spun down at 18,624 × g at 4 °C. The supernatants were measured for total protein concentrations and the samples were prepared accordingly in 4× NuPAGE LDS sample buffer. Subsequently the samples were run on NuPAGE™ 4–12% Bis-Tris gels (Invitrogen, CA, USA) and immunoblotted using the indicated antibodies.

### Immunofluorescence

Cells were seeded on 24-well glass coverslips and infected with ZIKV at MOI 5 and fixed with 4% PFA at indicated time points. The cells were subsequently permeabilised using 0.1% Triton X-100 for 5 min and blocked with 5% normal goat serum in 0.1% Triton X-100 for 1 h at room temperature. Cells were then probed with indicated antibodies. Nuclei were stained with either DAPI or Hoechst stain. The coverslips were mounted on glass slides and visualised using Carl Zeiss LSM confocal microscope. Images were analysed using ZEISS ZEN Microscope Software (ZEISS, Germany).

### TEM

Transmission Electron Microscopic (TEM) pictures were performed in samples prepared and analysed in the Electron Microscope Unit (Department of Pathology, HKU). Wildtype and Ube2g2$^{-/-}$ cells were seeded at a cell density of 1.5×10$^6$ in 10-cm dishes and infected with ZIKV at MOI 5. After 1-h incubation at 37 °C, the cells were washed and incubated in full DMEM medium (10% FBS) for 24 or 72 h. At each timepoint the cells were collected by gently scraping cells of the dishes and fixing in 1:10 v/v of 2.5% glutaraldehyde in 0.1 M sodium cacodulate-HCl (pH7.4) overnight at 4 °C. As controls, two samples were collected at 24 h that were subjected to control media rather than ZIKV.

### Pulse-chase analysis

Pulse chase experiments were performed as previously described[51,52]. Briefly, cells were trypsinised and starved for 30 min in methionine/cysteine free DMEM at 37 °C before pulse labelling. Cells were labelled for 10 min at 37 °C with 10 mCi/ml [$^{35}$S]methionine/cysteine (expressed protein mix; PerkinElmer) and chased for the indicated time points. At appropriate time points, aliquots were withdrawn and the reaction was stopped with cold PBS. Cell pellets were lysed in Tris buffer containing TX-100 and pre-cleared with agarose beads for 1 h at 4 °C. Immunoprecipitations were performed for 3 h at 4 °C with gentle agitation. Samples were eluted by heating in reducing or non-reducing sample buffer as indicated, subjected to SDS-PAGE and visualised by autoradiography.

### LD imaging

For quantitation of LDs using confocal microscope, WT and Ube2g2$^{-/-}$ cells were infected with ZIKV at MOI 5 and fixed at indicated time points.

The intracellular LDs were stained with Nile red dye, and the average positive Nile red area per cell was calculated using "Analyze Particles" macro in Image J software. The quantitation was performed from three independent experiments with a sample size of >500 cells for each condition used to calculate statistical significance. For quantitation using flow cytometry analysis, cells were infected with ZIKV at MOI 5. At indicated time points, cells were incubated with BODIPY™ 493/503 (Thermo Fisher, USA) for 30 min to stain intracellular LD, followed by fixation with 4% PFA for 30 min at RT. The analysis was performed with Attune NxT acoustic focusing cytometer (Thermo Fisher, MA, USA), and the percentage of BODIPY positive population and the mean fluorescence intensity compared to unstained controls were plotted.

### Fluorescence fatty acid labelling
HeLa-rUbe2g2 eGFP cells were first starved by culturing in HBSS media for 1 h. Cells were then incubated in culture medium containing 1 μM BODIPY® 558/568 C12 for 16 h prior to infection, followed by washing three times with PBS and subjected to virus infection. After 1 h of absorption, the cells were chased in HBSS buffer for a further 24 h and then fixed with 4% PFA. Cells were probed with anti-ZIKV capsid antibody and visualised by confocal imaging.

### Statistics and reproducibility
Data are presented as mean values ± SD (standard deviation) or SEM (standard error of mean), calculated using Microsoft Excel 2016/version 16.36 and GraphPad Prism version 9.3.1. $P$ value < 0.05 was considered the threshold for statistical significance. $P$ value significance intervals (*) are provided within each figure legend, together with the statistical test performed for each experiment. Significance was calculated in GraphPad Prism using one-way ANOVA followed by post hoc test for comparisons or unpaired Student's $t$ test. $N$ values are indicated within figure legends and refer to biological replicates. Derived statistics correspond to analysis of averaged values across biological replicates. Samples sizes were selected keeping in mind the variability between independent sources of cells. All experiments were performed in mammalian cell cultures, which are population-based, with data points generated from experiments performed from cells generated from independent clones and performed independent of each other. Sample sizes were determined based on the numbers required to achieve statistical significance using indicated statistics, but with a minimum of 3 independently performed experiments to ensure data reproducibility. No data was excluded from the analyses. The Investigators were not blinded to allocation during experiments and outcome assessment.

### Reporting summary
Further information on research design is available in the Nature Portfolio Reporting Summary linked to this article.

## Data availability
A reporting summary for this article is available as Supplementary Information file. Protein sequences used in this study were extracted from spectra that were searched against the human protein sequences in the Swiss-Prot database (database release version of May 2022), containing 20,621 sequences (www.uniprot.org) (Human; including isoforms and unreviewed sequences). Accession codes of RNAi experiments are NM_003262 for Sec62, NM_176812 for Chmp4b and NM_182688 for Ube2g2 with their corresponding catalogue number provided in the supplementary information. Data and immunoblots presented in the manuscript are provided as a Source Data file. Data generated or analysed during the current study are available through Figshare. Source data are provided with this paper.

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

## Acknowledgements
This work was supported by research grants from the Research Grants Council of the Hong Kong Special Administrative Region (17113019) (S.S.), Health and Medical Research Funds (19180912) (S.S.) and 17161032 (Y.L.) and the Wellcome Trust, UK (220776/Z/20/Z and 223107/Z/21/Z) (S.S.).

## Author contributions
Y.L., S.W.v.L., J.F., L.S., J.M.N., M.K., H.H.W., M.Y.L., J.Z., and S.S. performed and analysed experiments. Y.L. and S.W.v.L. helped write the manuscript. S.S. designed and conducted the study and wrote the manuscript.

## Competing interests
The authors declare no competing interests.
