## [Peer Review File · Nature Communications]

Viral subversion of selective autophagy is critical for biogenesis of virus replication organellesREVIEWER COMMENTS

Reviewer #1 (Remarks to the Author):

In this manuscript, Lan and co-authors, starting from a previous screen that identified AUP1 and Ube2g2 as dependency factors for flaviviruses, describe a role for the latter, an E2-ubiquitin ligase, in virus assembly and replication. Knockout of Ube2g2 impairs flaviviruses and coronaviruses infectivity and reconstitution of KO cells with wild type but not the catalytic inactive mutant was able to rescue the viral phenotype. The authors show that Ube2g2 might act on two different levels: on one hand loss of Ube2g2 impairs lipophagy and therefore inhibits viral particles assembly and release. On the other hand, Ube2g2, independently from its role in lipophagy, might contribute to ER membrane remodeling required for the formation of the viral replication compartments. Such role is performed by Ube2g2 mediated degradation of chaperones induced by viral-triggered UPR. The accumulation of selective chaperones upon loss of Ube2g2 induces Sec62/Chmp4 mediated ER-phagy, leading to the degradation of the non-structural proteins required for replication complex formation.

The experiments that the authors performed are technically sound, and certainly, the role of Ube2g2 in flavivirus life cycle and how Ube2g2 influences the regulation of ER homeostasis is of interest. However, there are several discrepancies throughout the manuscript that need further clarification.

It is somewhat counter-intuitive that in Ube2g2-KO infected cells, although there is a very robust reduction of viral replication (Figure 2e), paralleled by the almost disappearing of the NS proteins and more than 1-log reduction in infectivity, the level of the structural proteins is only mildly affected. Lack of viral replication will limit the amount of polyprotein and therefore the structural proteins should be reduced as well. How the authors explain this discrepancy? Which is the percent of infected Ube2g2-KO cells at the different time points? This will help to understand the magnitude of the effect.

According to the authors, Ube2g2 has a dual role during viral infection: it contributes to virus assembly/release through its function in the lipophagy pathway and it contributes to viral replication by regulating UPR-induced chaperone. The authors state that these two functions are independent yet regulated by the enzymatic activity. However, while fatty acid reconstitution cannot rescue the assembly/release phenotype, knock-down of Chmp4/Sec62 can completely rescue the viral phenotype in Ube2g2-KO cells. However, Chmp4/Sec62 KO should not compensate the defect in assembly/release. Thus, how much Ube2g2-associated lipophagy contributes to the viral phenotype?

These and some additional concerns (see specific comments) must be addressed to strengthen the authors' results and consolidate their claims.

Specific comments:

Figure 4: The authors showed that VLP secretion as well as VSVG trafficking is impaired in Ube2g2 KO cells. While the authors showed that lipophagy is impaired in Ube2g2 cells, it is not clear why lipophagy should be the main reason for the trafficking/assembly defect. Specifically, there are no reports that show that VLPs production relies on lipophagy, or lipophagy is activated during VLP production. Thus, to draw these conclusions the authors should show that indeed lipophagy is activated during ZIKV VLPs production (do lipid droplets decrease in size and number during VLP production?).

Figure 6

a: Why in the infected AupKO cells there is no Ube2g2 signal?

e: please show a marker for viral infection (E or dsRNA)

Again in this case, it appears that all the cells in the Ube2g2-KO cells show a marked phenotype for both autophagy flux. Are all the cells infected? Should not be the case considering the reduction in replication and assembly/release/spread.

Figure 7d: The authors should include a viral protein in the WB such as NS3 since it is the one mediating the FAM134b cleavage

Line 330: reference is not completely accurate since it describes activation of UPR to mitigate flavivirus induced cytopathic effect but not membrane proliferation as written by the authors

The title describes as if virus triggers inhibition of ER-phagy. This is certainly true for FAM134b

mediated ER-phagy, in which viral proteins have an active role in restricting the pathway. In this manuscript there is no indication that viral infection modulates Ube2g2 activity, either by increasing its level or function. Rather, the loss of Ube2g2 allows for ER-phagy to occur and thus blocking viral replication. The title might need some rephrasing to better match the author's findings

Fig5 a-d: In the figure legends the authors write about average Nile red area, however there is not such analysis in the figure. Please clarify.

Fig5c: infection marker missing

Fig6d: The merge panels for Mock cells are black.

Figure 7: which is the difference between figure 7b and 7c?

Reviewer #2 (Remarks to the Author):

The manuscript entitled "Viral inhibition of ER-phagy is critical to membrane remodelling for biogenesis of virus replication organelles" investigates the effect of Ube2g2 loss upon infection by (+)RNA viruses (flaviviruses and coronaviruses). The authors proposed the role of Ube2g2 regarding viral replication. Their hypothesis is that Ube2g2 is important to promote lipophagy and at the same time it negatively regulates ER-phagy. Both forms of selective autophagy affect viral replication. Authors proposed that reduction of ER-phagy accompanied with induction of lipophagy drives host membrane rewire to allow biogenesis of viral replication organelles. The role of ER-phagy and Lipophagy in viral infection and viral proliferation is known. The role of Ube2g2 is a novelty.

The biggest problem of the manuscript is that hypothesis are not properly supported by scientific data. While the viral infection and proliferation assays are fairly convincing, as well as the related biochemistry, the autophagy and ER remodelling part are poor in terms of scientific outcomes and data presentation.

1. Lipophagy assays are missing. Authors simply quantify the amount of lipid droplets (LD) however there are not evidences that the LD are degraded into lysosomes. High resolution microscopy and/or electron microscopy are needed.
2. Macro-autophagy assays are limited to LC3B-II WB and the use of the tandem GFP-RFP-LC3B. In principle, these two assays are correct but the quality of images is too low. Moreover, the analysis of autophagosome using EM is not explicative. Authors should consider to analyse autophagy induction following the indications provided in many review articles (e.i. Klionsky et al., Autophagy)
3. ER-phagy is poorly addressed only at the end of the manuscript. The title of the manuscript is clearly pointing on ER-phagy. The role of ER-phagy during viral infection has been addressed in the past. There are several tools and reagents that can be used. The authors limited their analysis to Sec62 that is just one out of the eight ER-phagy receptors described so far. Same as for lipophagy, there are clear experiments showing that ER portions are delivered or not to lysosomes.
4. ER morphology has not been addressed in a proper way. High resolution microscopy and or more clear EM images are needed to clearly show that the ER is changing its morphology. Will be useful to investigate the ER proteins amount and the type of ER proteins that are involved in ER remodelling during infections.

Even if the Ube2g2 is an interesting point, the manuscript needs a significant improvement before been considered for publication

Reviewer #3 (Remarks to the Author):

Expansion and remodeling of the ER is fundamental to establishment of replication organelles by flaviviruses and coronaviruses and therefore important to understand. The authors previously identified lipid droplet associated protein Ancient ubiquitin protein 1 (Aup1) and the ubiquitin

conjugating protein Ube2g2 as host dependency factors, but the function of Ube2g2 was not characterized. Here, Ube2g2 was shown to be required for flavivirus and coronavirus replication at a post-entry step. In the absence of active Ube2g2, expression of flavivirus nonstructural proteins was reduced, most likely due to their rapid degradation, which in turn resulted in a failure of virus replication to establish replication organelles in the ER. Ube2g2-deficient cells were impaired in lipophagy, and the defects could be recovered by depleting Sec62 and Chmp4, demonstrating that Ube2g2 inhibits ER-phagy to facilitate membrane expansion and remodeling.

This work represents an important insight into biogenesis of the replication compartments for important viruses, but a number of clarifications in the data and methods are needed.

Specific comments:

1. The specific cell type is not mentioned in the main text but appears to be HeLa cells. This needs to be justified as HeLa's have no biological relevance to flavivirus or coronavirus replication. In addition, the statistical analysis used is not mentioned anywhere and needs to be included. No methods are included or validation data for the replicon are included and should be shown. The source and identification of the antibodies used in the study are not included. These oversights in methods need to be addressed.
2. Figure 1: Cells reconstituted with Ube2g2 have considerably lower gapdh expression raising the question of how cell viability impacts these findings. Figure 1c shows lower ubiquitination, but the loading controls are also considerably lower. Therefore, measurements of cell viability during virus infection (Figure 1d) need to be shown to verify that the effects observed are not due to cell death. In addition, inclusion of a control virus that replicates in the nucleus (eg influenza or HSV) is also needed to show the specificity of Ube2g2 function to ER medication by positive strand viruses.
3. Figure 2f: RNA /replicon replication in the Ube2g2 reconstituted cells (wt and catalytic mutant) should be shown here to establish that effects of the host enzyme are also only at the level of RNA replication and not RNA packaging. The results suggest that flavivirus replication likely recruits Ube2g2 to inhibit ER-phagy locally – is this observed with the replicon with localization of Ube2g2 together with dsRNA?
4. The conclusions from Figure 6 d and e are not clear. LAMP2 staining doesn't look different between the WT and Ube2g2^{-/-} cells and any differences seem to be a function of the number of cells per field. In addition, the mock cells expressing LC3-GFP-RFP in panels d and e looks vastly different to each other without any interpretation. This data needs improvement.
5. Figure 7d: The specificity of antibodies to detect cleavage products of FAM134B and Herpud1 need to be verified using siRNAs. Lines 287-288 don't seem to reflect the data where FAM134B cleavage is reduced associated with reduced nonstructural protein expression?
6. Figure 7i: blots for the ER-stress response proteins need to be shown as they are important to the conclusion here.

Minor comments:

- dengue virus is not capitalized as the virus is not named after a place (unlike Zika virus).
- Figure 3c – what time point was sampled to examine production of virus particles?
- Line 206 – pr and M are not separate proteins
- Figure 6a – it should be noted that loss of Aup1 results in loss of Ube2g2

Reviewer #1 (Remarks to the Author):

In this manuscript, Lan and co-authors, starting from a previous screen that identified AUP1 and Ube2g2 as dependency factors for flaviviruses, describe a role for the latter, an E2-ubiquitin ligase, in virus assembly and replication. Knockout of Ube2g2 impairs flaviviruses and coronaviruses infectivity and reconstitution of KO cells with wild type but not the catalytic inactive mutant was able to rescue the viral phenotype. The authors show that Ube2g2 might act on two different levels: on one hand loss of Ube2g2 impairs lipophagy and therefore inhibits viral particles assembly and release. On the other hand, Ube2g2, independently from its role in lipophagy, might contribute to ER membrane remodeling required for the formation of the viral replication compartments. Such role is performed by Ube2g2 mediated degradation of chaperones induced by viral-triggered UPR. The accumulation of selective chaperones upon loss of Ube2g2 induces Sec62/Chmp4 mediated ER-phagy, leading to the degradation of the non-structural proteins required for replication complex formation. The experiments that the authors performed are technically sound, and certainly, the role of Ube2g2 in flavivirus life cycle and how Ube2g2 influences the regulation of ER homeostasis is of interest. However, there are several discrepancies that need further clarification.

It is somewhat counter-intuitive that in Ube2g2-KO infected cells, although there is a very robust reduction of viral replication (Figure 2e), paralleled by the almost disappearing of the NS proteins and more than 1-log reduction in infectivity, the level of the structural proteins is only mildly affected. Lack of viral replication will limit the amount of polyprotein and therefore the structural proteins should be reduced as well. How the authors explain this discrepancy? Which is the percent of infected Ube2g2-KO cells at the different time points? This will help to understand the magnitude of the effect.

Author response: We thank the reviewer for raising this. This is indeed what we observe in a multicycle replication. At earlier timepoints, 12 and 24 hours (single cycle of replication), infection in the WT and KO cells is equivalent and the expression levels of the structural proteins in the KO cells is comparable to that of the WT indicating that translation is not affected. However, at 48 h and later time points there is a substantial reduction in the steady state levels of structural proteins too (Figure 3a; also evident in Figure 2g), in line with the fact that there is limited viral RNA and therefore reduced polyprotein in the later timepoints, as the reviewer points out. We have clarified this in the revised manuscript.

According to the authors, Ube2g2 has a dual role during viral infection: it contributes to virus assembly/release through its function in the lipophagy pathway and it contributes to viral replication by regulating UPR-induced chaperone. The authors state that these two functions are independent yet regulated by the enzymatic activity. However, while fatty acid reconstitution cannot rescue the assembly/release phenotype, knock-down of Chmp4/Sec62 can completely rescue the viral phenotype in Ube2g2-KO cells. However, Chmp4/Sec62 KO should not compensate the defect in assembly/release. Thus, how much Ube2g2-associated lipophagy contributes to the viral phenotype?

Author response: We thank the reviewer for pointing this out. This was an omission in the description of the methods – the depletion of Chmp4/Sec62 was done in media supplemented with 10 mM free fatty acids to circumvent the defect in lipophagy, based on our data from Figure 6. We have clarified this in the revised manuscript.

These and some additional concerns (see specific comments) must be addressed to strengthen the authors' results and consolidate their claims.

Specific comments:

Figure 4: The authors showed that VLP secretion as well as VSVG trafficking is impaired in

Ube2g2 KO cells. While the authors showed that lipophagy is impaired in Ube2g2 cells, it is not clear why lipophagy should be the main reason for the trafficking/assembly defect. Specifically, there are no reports that show that VLPs production relies on lipophagy, or lipophagy is activated during VLP production. Thus, to draw these conclusions the authors should show that indeed lipophagy is activated during ZIKV VLPs production (do lipid droplets decrease in size and number during VLP production?).

Author response: We thank the reviewer for raising this. Contribution of lipid droplet turnover to specific cargo, e.g. VSV-G secretion has been reported previously (Tapia D et al, 2019 – traffic induced degradation response for secretion, Simpson et al, 2012). We have also provided additional data showing decreased lipid droplet numbers in VLP producing cells in the revised manuscript (revised Figure 4d, e).

Figure 6a: Why in the infected AupKO cells there is no Ube2g2 signal?

Author response: Aup1 has been shown to bind to and stabilize Ube2g2 (Smith CE et al, 2021); therefore, in the absence of Aup1 at later timepoints in infection the signal from Ube2g2 disappears. We have modified the text in the revised manuscript to explain this.

Figure 6e: please show a marker for viral infection (E or dsRNA)

Again, in this case, it appears that all the cells in the Ube2g2-KO cells show a marked phenotype for both autophagy flux. Are all the cells infected? Should not be the case considering the reduction in replication and assembly/release/spread.

Author response: We thank the reviewer for raising this. The induction in autophagic flux occurs in Ube2g2 KO cells irrespective of infection, i.e., we see this phenotype also in mock infected cells as seen biochemically too (Figure 6b); this becomes more pronounced upon infection. We believe that inhibition of basal ER-phagy is maintained by Ube2g2 activity, but becomes acutely necessary upon infection. Loss of Ube2g2 therefore leads to widespread induction of ER-phagy.

Including a viral marker in 6e is difficult after using the GFP, RFP and magenta channels for LC3 and Lamp2. We therefore have the viral marker included in Fig 6d (which is essentially the same expt) to visualise the extent of infection.

Figure 7d: The authors should include a viral protein in the WB

Author response: We thank the reviewer for raising this. We have included a viral marker in the revised manuscript (new Figure 7d).

Line 330: reference is not completely accurate since it describes activation of UPR to mitigate flavivirus induced cytopathic effect but not membrane proliferation as written by the authors.

Author response: We thank the reviewer for pointing this out. We have modified the text to correct this error.

The title describes as if virus triggers inhibition of ER-phagy. This is certainly true for FAM134b mediated ER-phagy, in which viral proteins have an active role in restricting the pathway. In this manuscript there is no indication that viral infection modulates Ube2g2 activity, either by increasing its level or function. Rather, the loss of Ube2g2 allows for ER-phagy to occur and thus blocking viral replication. The title might need some rephrasing to better match the author's findings

Author response: We believe that viral NS3 protease potentially generates substrates for Ube2g2-dependent degradation (Figure 7i), which results in impaired ER-phagy.

Fig 5 a-d: In the figure legends the authors write about average Nile red area, however there is not such analysis in the figure. Please clarify.

Author response: We have corrected this error and included the quantification in the revised manuscript.

Fig 5c: infection marker missing

Author response: We have included the infection marker for Fig 5c.

Fig 6d: The merge panels for Mock cells are black.

Author response: We have corrected this error.

Figure 7: which is the difference between figure 7b and 7c?

Author response: Panel b indicates the fold change of only the spliced form of Xbp1 while panel c indicates total Xbp1 transcripts, as indicated in the y axes.

Reviewer #2 (Remarks to the Author):

The manuscript entitled “Viral inhibition of ER-phagy is critical to membrane remodelling for biogenesis of virus replication organelles” investigates the effect of Ube2g2 loss upon infection by (+)RNA viruses (flaviviruses and coronaviruses). The authors proposed the role of Ube2g2 regarding viral replication. Their hypothesis is that Ube2g2 is important to promote lipophagy and at the same time it negatively regulates ER-phagy. Both forms of selective autophagy affect viral replication. Authors proposed that reduction of ER-phagy accompanied with induction of lipophagy drives host membrane rewire to allow biogenesis of viral replication organelles. The role of ERphagy and Lipophagy in viral infection and viral proliferation is known. The role of Ube2g2 is a novelty.

While the viral infection and proliferation assays are fairly convincing, as well as the related biochemistry, the autophagy and ER remodelling part are poor in terms of scientific outcomes and data presentation.

1. Lipophagy assays are missing. Authors simply quantify the amount of lipid droplets (LD) however there are not evidences that the LD are degraded into lysosomes. High resolution microscopy and/or electron microscopy are needed.

Author response: We thank the reviewer for suggesting this. The involvement of lipophagy is based on our previous work (Zhang et al, 2018) where we have characterised the involvement of Aup1 in virus-triggered lipophagy. However, we have added microscopy data of LDs in the presence and absence of Bafilomycin to show lysosomal colocalization and turnover of LDs, in the revised manuscript (new extended Figure 3c, d).

2. Macro-autophagy assays are limited to LC3B-II WB and the use of the tandem GFP-RFP-LC3B. In principle, these two assays are correct but the quality of images is too low. Moreover, the analysis of autophagosome using EM is not explicative. Authors should consider to analyse autophagy induction following the indications provided in many review articles (e.i. Klionsky et al., Autophagy)

Author response: We have included additional microscopy data using GFP-LC3 and Lamp2 (as one of the indications of autophagy induction suggested in Klionsky et al) as a time-course of infection to corroborate our findings on induction of autophagosomes (new extended Figure

4c). Infected cells show induction of LC3 puncta and their colocalization with Lamp2, specifically in the infected cells.

ER-phagy is poorly addressed only at the end of the manuscript. The title of the manuscript is clearly pointing on ER-phagy. The role of ER-phagy during viral infection has been addressed in the past. There are several tools and reagents that can be used. The authors limited their analysis to Sec62 that is just one out of the eight ER-phagy receptors described so far. Same as for lipophagy, there are clear experiments showing that ER portions are delivered or not to lysosomes.

Author response: We thank the reviewer for raising this. We have measured expression of six ER-phagy receptors (new Figure 7) - we could not get antibodies to CCPG1 and Tex264 to work in our cells - and the only one which shows altered expression in the Ube2g2 KO cells is Sec62. Interestingly, ATL3, which has been shown to be important for ER remodelling during flavivirus infection was found to be degraded in the Ube2g2-deficient cells, further confirming our model. We therefore specifically analysed Sec62 in further detail. We have included additional microscopy images from wild-type and Ube2g2-deleted cells, showing Sec62 and Lamp1 to showing the typical ER-phagy morphology as supporting data (new extended Fig 5).

4. ER morphology has not been addressed in a proper way. High resolution microscopy or more clear EM images are needed to clearly show that the ER is changing its morphology. Will be useful to investigate the ER proteins amount and the type of ER proteins that are involved in ER remodelling during infections.

Author response: We thank the reviewer for raising this point. We have included microscopy data for the ER in the revised manuscript (new extended Fig 5d) showing the typical morphology of ER surrounded by Lamp1 positive compartments in Ube2g2^{-/-} cells. We have also included Atlastins (ATL3) as a ER-remodelling protein, which appears to undergo degradation in the Ube2g2^{-/-} cells (new Figure 7).

Reviewer #3 (Remarks to the Author):

Expansion and remodelling of the ER is fundamental to establishment of replication organelles by flaviviruses and coronaviruses and therefore important to understand. The authors previously identified lipid droplet associated protein Ancient ubiquitin protein 1 (Aup1) and the ubiquitin conjugating protein Ube2g2 as host dependency factors, but the function of Ube2g2 was not characterized. Here, Ube2g2 was shown to be required for flavivirus and coronavirus replication at a post-entry step. In the absence of active Ube2g2, expression of flavivirus nonstructural proteins was reduced, most likely due to their rapid degradation, which in turn resulted in a failure of virus replication to establish replication organelles in the ER. Ube2g2-deficient cells were impaired in lipophagy, and the defects could be recovered by depleting Sec62 and Chmp4, demonstrating that Ube2g2 inhibits ER-phagy to facilitate membrane expansion and remodeling.

This work represents an important insight into biogenesis of the replication compartments for important viruses, but a number of clarifications in the data and methods are needed.

Author response: We thank the reviewer for their favourable view of our findings.

Specific comments:

1. The specific cell type is not mentioned in the main text but appears to be HeLa cells. This is needs to be justified as HeLa's have no biological relevant to flavi or corona-virus replication.

In addition, the statistical analysis used is not mentioned anywhere and needs to be included. The source and identification of the antibodies used in the study are not included. These oversights in methods need to be addressed.

Author response: We thank the reviewer for raising these points. We have previously validated both Dengue and Zika infections performed in HeLa cells, in hepatocyte cell lines and primary hepatocytes (Zhang et al, 2018, Li et al, 2020). We have therefore used HeLa cells for these studies. We have however included results from Huh7 hepatocytes (new extended Figure 1c), which confirms the loss of replication upon Ube2g2-deletion that we observe in HeLa cells.

We have included details of methods, statistical analyses as well as sources and catalogue nos of antibodies in the revised manuscript and the reporting checklist respectively.

2. Figure 1: Cells reconstituted with Ube2g2 have considerably lower gapdh expression raising the question of how cell viability impacts these findings. Figure 1c shows lower ubiquitination, but the loading controls are also considerably lower. Therefore, measurements of cell viability during virus infection (Figure 1d) need to be shown to verify that the effects observed are not due to cell death. In addition, inclusion of a control virus that replicates in the nucleus (eg influenza or HSV) is also needed to show the specificity of Ube2g2 function to ER medication by positive strand viruses.

Author response: These are indeed important points. We have included cell viability assays (new extended Fig 1a) and influenza infection as control virus (new extended Fig 1b) in the revised manuscript as per the reviewer's suggestions.

3. Figure 2f: RNA /replicon replication in the Ube2g2 reconstituted cells (wt and catalytic mutant) should be shown here to establish that effects of the host enzyme are also only at the level of RNA replication and not RNA packaging. The results suggest that flavivirus replication likely recruits Ube2g2 to inhibit ER-phagy locally – is this observed with the replicon with localization of Ube2g2 together with dsRNA?

Author response: We thank the reviewer for raising this point. We have included replicon results from the cells reconstituted with the wild-type and catalytically dead Ube2g2 (new Fig 2f) in the revised manuscript.

With regards to specifically visualising Ube2g2 with dsRNA, this has proven to be technically very difficult to achieve, on account of the dual localisation of Ube2g2 with that of lipid droplets and the ER. Even with the replicon, it appears in both the ER and LDs are therefore difficult to visualise specifically with dsRNA.

4. The conclusions from Figure 6 d and e are not clear. LAMP2 staining doesn't look different between the WT and Ube2g2^{-/-} cells and any differences seem to be a function of the number of cells per field. In addition, the mock cells expressing LC3-GFP-RFP in panels d and e looks vastly different to each other without any interpretation. This data needs improvement.

Author response: We have included higher resolution images for Figs 6d and 6e. The abundance of the autophagosomes and autolysosomes in the mock cells typically varies based on the basal conditions (cell passages, and morphology), and therefore the GFP and RFP signals can be significantly different.

5. Figure 7d: The specificity of antibodies to detect cleavage products of FAM134B and Herpud1 need to be verified using siRNAs. Lines 287-288 don't seem to reflect the data where FAM134B cleavage is reduced associated with reduced non-structural protein expression?

Author response: we have included siRNA-based verification of FAM134B and Herpud1 using siRNAs (new extended Fig 6a and 6b). In virus-infected wild-type cells, we observe an

increased induction of total FAM134B over time, which appears in the cleaved form, whereas this is substantially lower in the Ube2g2-deficient cells, both at total levels and cleaved forms.

6. Figure 7i: blots for the ER-stress response proteins need to be shown as they are important to the conclusion here.

Author response: we have included the autoradiograms in the revised manuscript (new extended Figure 6b, c)

Minor comments:

- dengue virus is not capitalized as the virus is not named after a place (unlike Zika virus).

Author response: We have corrected this error.

- Figure 3c – what time point was sampled to examine production of virus particles?

Author response: 48 hours; we have included this detail in the revised manuscript.

- Line 206 – pr and M are not separate proteins

Author response: We have corrected this error.

- Figure 6a – it should be noted that loss of Aup1 results in loss of Ube2g2

Author response: We have clarified this point.

REVIEWERS' COMMENTS

Reviewer #1 (Remarks to the Author):

The authors have addressed all the points raised during the revision process. I do not have any additional concerns and thereby I support the manuscript publication.

Reviewer #2 (Remarks to the Author):

In the revised version of the manuscript, authors addressed the majority of the Reviewers' questions. The manuscript now is more clear and complete. However, the ER morphology and ER-phagy flux assays are still weak.

Reviewer #3 (Remarks to the Author):

The authors have addressed the comments of this reviewer to a satisfactory level. Thank you.